# Inhibition of polymorphonuclear cells averts cytotoxicity against hypoimmune cells in xenotransplantation

Xiaomeng Hu[1,2], Grigol Tediashvili[1], Alessia Gravina [1], Jonathan Stoddard[3], Trevor J. McGill[3], Andrew J. Connolly[4], Tobias Deuse [1,7] & Sonja Schrepfer [1,2,5,6,7] ✉

Allogeneic, immune-evasive hypoimmune (HIP) cell therapeutics that are HLA-depleted and overexpress CD47 create the opportunity to treat immuno-competent patients with cancer, degenerative, or autoimmune diseases. However, HIP cell therapy has not yet been established for xenotransplantation. Here we engineer, for human-to-non-human primate studies, human HIP* endothelial cells (EC) that are HLA-depleted and express macaque CD47 to allow compatibility with the macaque SIRPα immune checkpoint. Although no T cell, NK cell, or macrophage responses and no antibody-dependent cyto-toxicity is observed in cynomolgus recipients, we reveal that macaque poly-morphonuclear cells (PMN) show strong xenogeneic cytotoxicity against HIP* ECs. Inhibition of PMN killing using a multi-drug regimen leads to improved xenogeneic human HIP* EC survival in cynomolgus monkeys. Similarly, human PMNs show xenoreactivity against pig ECs, which has implications for clinical xenotransplantation. Accordingly, our engineered pig HIP* ECs that are SLA-depleted, overexpress human CD47, and additionally overexpress the PMN-inhibitory ligands CD99 and CD200, are protected against all human adaptive and innate cytotoxicity, including PMNs. In summary, specific targeting of PMN-mediated killing of the transplanted cells might improve outcomes for clinical pig-to-human xenotransplantation.

Cell replacement therapy builds on the principles that cell products can be manufactured and transplanted, will engraft and survive in host tissue, and exert persistent functions to overcome diseases. While autologous cell sources can establish proof of concept and produce therapeutics for some individuals, only large-scale allogeneic or xenogeneic manufacturing can produce cost-effective products that meet high-quality standards and supply larger patient populations. However, this approach requires engineering of the cells to become immune evasive so they can be transplanted into all patients, regardless of their HLA type or sensitization status, and without immunosuppression or supportive medication. Our hypoimmune (HIP) concept of inactivating both the *B2M* and *CIITA* genes, which leads to HLA class I- and II-deficiency, and overexpression of CD47 has so far shown great efficacy in several preclinical models[1–4]. Recent proof of concept studies in preclinical models showed that specialized allogeneic HIP cell products can be used to treat cardiovascular and pulmonary diseases[5], diabetes mellitus[6], and cancer[7].

[1]Department of Surgery, Division of Cardiothoracic Surgery, Transplant and Stem Cell Immunobiology (TSI)-Lab, University of California San Francisco, San Francisco, CA, USA. [2]Sana Biotechnology Inc., South San Francisco, CA, USA. [3]Division of Neuroscience, Oregon National Primate Research Center, Oregon Health & Science University, Portland, OR, USA. [4]Department of Pathology, University of California San Francisco, San Francisco, CA, USA. [5]Department of Surgery, Cedars-Sinai Medical Center, Los Angeles, CA, USA. [6]Board of Governors Regenerative Medicine Institute, Cedars-Sinai Medical Center, Los Angeles, CA, USA. [7]These authors contributed equally: Tobias Deuse, Sonja Schrepfer. ✉e-mail: Sonja.Schrepfer@cshs.org

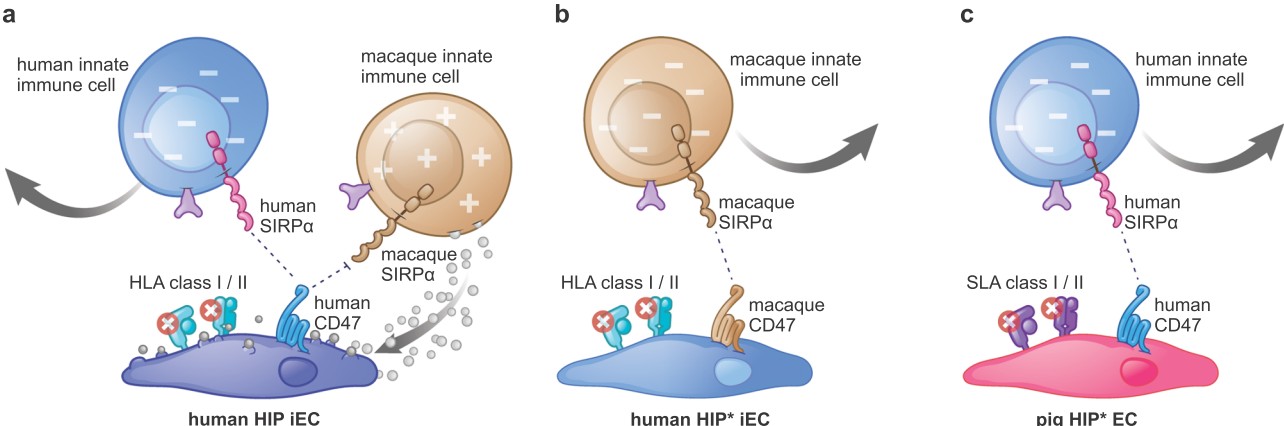

**Fig. 1 | The engineering of human HIP* and pig HIP* endothelial cells. a** Human HIP iECs (*B2M*⁻/⁻ *CIITA*⁻/⁻ human CD47-high) are protected from human NK cells and macrophages (innate immune cells) because human CD47 activates the immune checkpoint SIRPα. However, human CD47 does not interact effectively with macaque SIRPα, and this left HIP cells susceptible to killing by macaque innate immune cells. **b** Human HIP* iECs (*B2M*⁻/⁻ *CIITA*⁻/⁻ macaque CD47-high) are protected from macaque innate immune cells. Although macaque NK cells and macrophages sense HLA deficiency, the macaque CD47 interacts with macaque SIRPα and prevents innate cytotoxicity. **c** Pig HIP* ECs (*B2m*⁻/⁻ *Ciita*⁻/⁻ human CD47-high) are protected from human innate immune cells. Human NK cells and macrophages similarly sense SLA deficiency on pig cells, but the human CD47 protects them from being killed.

Xenotransplantation crosses a more stringent immunological barrier with more layers of rejection mechanisms. The aim of this study was to test our HIP concept in xenotransplantation. We chose endothelial cells (EC) as the cell type for human-to-non-human primate (NHP) in vivo xenotransplant studies and pig-to-human in vitro studies because ECs are particularly immunogenic and will maximize the breadth of immune activation of recipient immune cells. ECs derive their strong immunogenicity from being non-professional antigen-presenting cells expressing various co-stimulatory molecules[8,9]. ECs further express a large variety of heavily glycosylated adhesion molecules involved in leukocyte and neutrophil trafficking[10] and mediate transendothelial migration of immune cells. They form the interface between solid organ grafts and the xenogeneic immune system, and most described xenorejection phenomena involve the EC layer[11]. The results derived from this study are thus of relevance for cell xenotransplantation and solid organ xenotransplantation.

Transgenes expressed on cellular grafts must be of the recipient species to properly interact with the recipient immune system or have to be sufficiently cross-reactive to be functionally competent. However, using non-human analogs could introduce additional variables into the human cell product, and other groups have chosen to deprioritize mechanistic considerations for the benefit of testing unmodified human cells in NHPs[12,13]. We have recently identified species-specific differences in the CD47-SIRPα axis between humans and macaques[4]. Human HIP iECs were protected from human NK cell and macrophage killing but were expeditiously killed by their analogous rhesus macaque immune cells (Fig. 1a). Human HIP cells are therefore not protected from immune rejection in NHPs. For use with NHP recipients, we modified our HIP concept to include the macaque CD47 transgene instead of human CD47 (referred to as HIP*, Fig. 1b). Human HIP* iECs were previously shown to be protected from macaque NK cell and macrophage killing in vitro[4] and in vivo[1] and are suitable for NHP xenotransplant experiments. Similarly, for pig-to-human xenotransplant studies, we used pig HIP* ECs with depleted swine leukocyte antigen (SLA) class I and II and human CD47 transgene expression (Fig. 1c).

This study reveals that polymorphonuclear cells (PMN) pose an additional immune barrier for the xenotransplantation of HIP cells. We show that pharmacologic inhibition of PMNs or the overexpression of the PMN-inhibitory ligands CD99 and CD200 prevent PMN cytotoxicity. Pig HIP* cells expressing CD99 and CD200 became fully protected against all human adaptive and innate immune cell cytotoxicity including PMNs.

## Results

### ABO-mismatched ECs are subject to blood type antibody-mediated killing

The necessity for ABO blood group matching for cell transplantation is not well studied and understood. We therefore performed in vitro impedance cytotoxicity assays with human ECs from donors of all major ABO blood groups. The target cells were plated on microtiter plates containing interdigitated gold microelectrodes, and their viability was assessed using electrical impedance as the readout. A drop of the normalized cell index (CI) curve indicates target cell damage and death. The ECs were incubated with serum from human donors of different blood groups in complement-dependent cytotoxicity (CDC) assays (Fig. 2a). We observed rapid killing in all the configurations that are knowingly avoided for blood transfusions but saw unimpeded target cell survival in all configurations that are safe for blood transfusions. We next tested whether human serum containing anti-Rh(D) antibodies could exert cytotoxicity against human Rh(D)+ target cells (Fig. 2b). Rh(D)-expressing ECs of the blood type A, B, and O were all killed in ABO-identical serum containing anti-Rh(D) antibodies. In contrast, Rh(D)-negative ECs remained unaffected by anti-Rh(D) antibodies (Fig. 2c). The human HIP* iECs used in this study were blood type A Rh(D)+. When incubated with human serum of all the main blood types, they were killed rapidly in human serum of blood type B and O (Fig. 2d). Human HIP* iECs were also susceptible to killing mediated by anti-Rh(D) antibodies in blood type-matched serum (Fig. 2e). In preparation for our NHP experiments, we tested human HIP* iECs with rhesus macaque serum of blood type B, which is the most prevalent blood type among rhesus monkeys, while blood type A and AB are rare. We observed rapid killing in this xenotransplant setting in vitro (Fig. 2f). These results showed that human ECs cannot be transplanted in an ABO blood type-incompatible manner into NHPs. This killing is independent of HIP* editing and affects both primary ECs and iECs.

### Human HIP* iEC grafts are subject to PMN-mediated killing in an ABO-matched rhesus monkey

After screening two primate colonies, we identified only one compatible rhesus monkey with blood type AB that was available for our study. We injected human HIP* iEC grafts and performed a comprehensive immune analysis with immune cells and serum isolated before and 7 days after the cell injection (Fig. 3a). In this study, we tested all recipient immune cells, including polymorphonuclear cells (PMN), a major immune cell population that

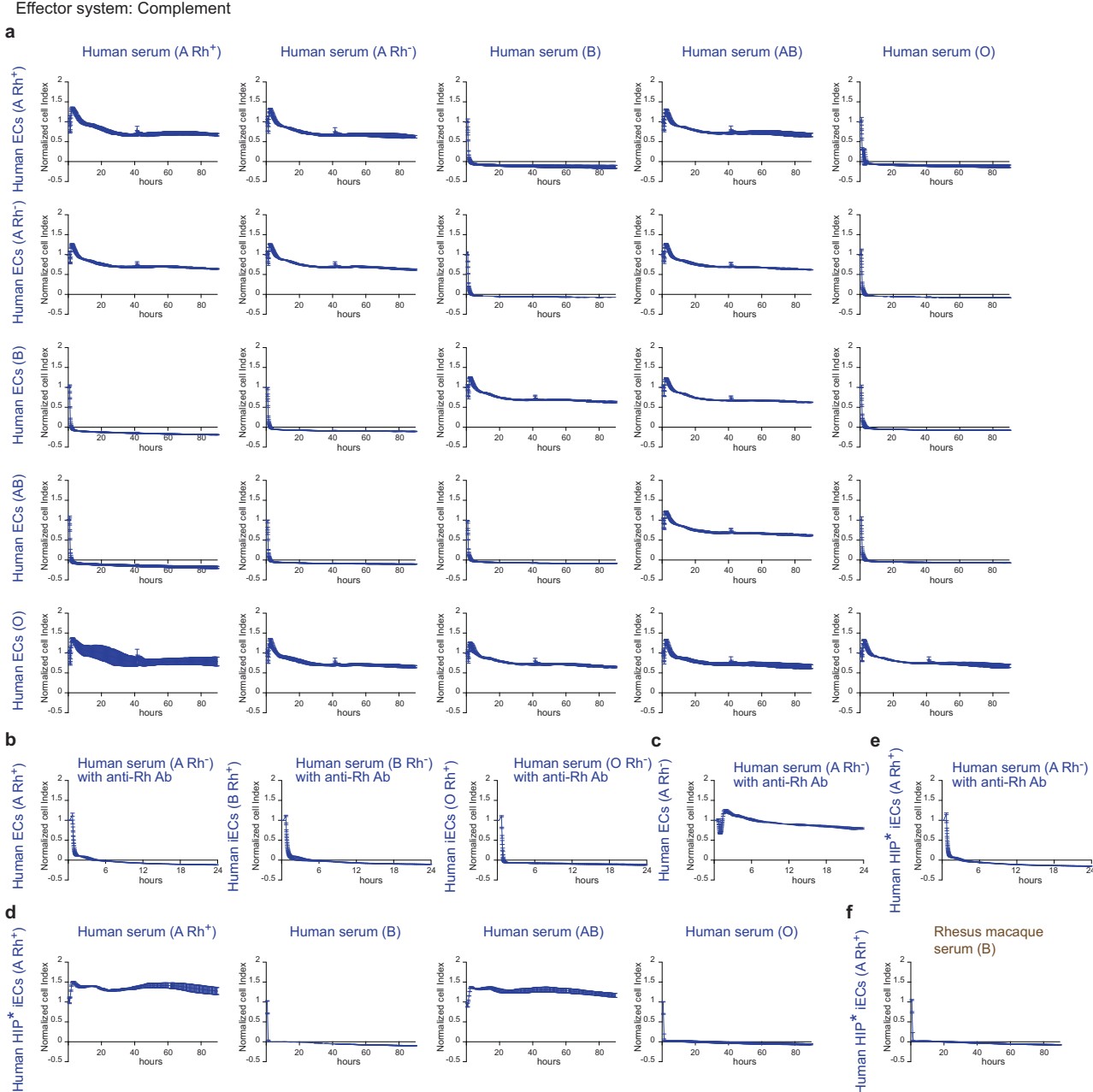

**Fig. 2 | ABO-mismatched human and rhesus macaque sera kill human endothelial cells. a** Human endothelial cells (ECs) from donors of all major ABO blood groups were incubated with serum from donors with different blood groups in CDC assays and cytotoxicity was assessed in impedance assays. ECs were killed by all ABO blood type-incompatible sera. **b** Rh(D)-expressing ECs of the blood type A, B, and O were incubated with blood type-identical serum containing anti Rh(D) antibodies. All ECs were killed in CDC assays. **c** Rh(D)-negative ECs of the blood type A were incubated with blood type A serum containing anti Rh(D) antibodies. No cytotoxicity was observed. **d** Human HIP* iECs of the blood type A Rh(D)+ were

incubated with human serum of all the main blood types not containing Rh(D) antibodies. Expectedly, HIP* iECs were killed only by blood type B and O sera, which contain anti-A antibodies. **e** When human HIP* iECs were incubated with blood type-identical human A serum that contained Rh(D) antibodies, they were killed. **f** In a xeno setting, human HIP* iECs were incubated with rhesus macaque serum of blood type B and were killed. In this Figure, all human cells or serum are labeled in blue and rhesus macaque serum in brown. Data are shown as mean ± SD in technical triplicates.

is not often assessed in transplant studies. Immune cells were separated into PMNs, T cells, NK cells, and macrophages. The percent drop in CI at 90 h on impedance cytotoxicity assays was recorded. Negative values reflect an increase in CI from proliferation or change in cell morphology, while values > 100 reflect increased conductivity from cellular electrolytes after complete disintegration of the target cells. The assays showed no killing activity by rhesus monkey T cells, NK cells, or macrophages and no cytotoxicity via ADCC or CDC (Fig. 3b). Therefore, these

results confirmed the data obtained in a similar study previously[1]. However, the human HIP* iECs were rapidly killed by rhesus macaque PMNs from pre- and post-transplant blood (Fig. 3c). The fact that even pre-transplant PMNs were strongly reactive against the HIP* iECs pointed towards an innate, xeno-specific activity that does not require sensitization. Additionally, the absence of killing in ADCC and CDC assays showed that this was not an antibody-mediated killing. The data mechanistically suggested direct cellular immune recognition by PMNs.

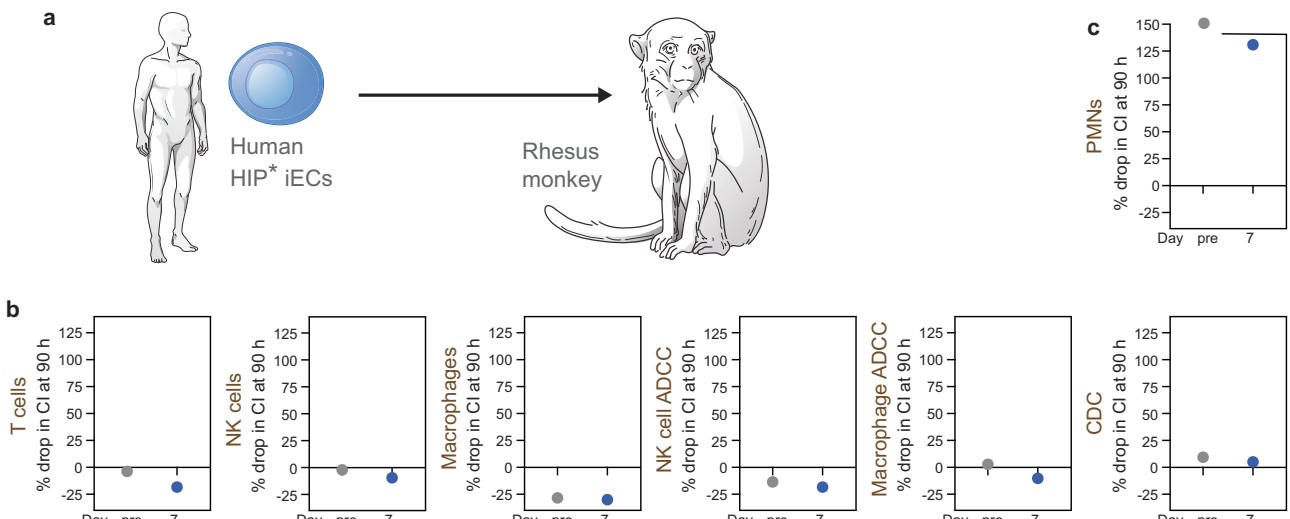

**Fig. 3 | PMNs are the only immune cell population reactive against human HIP\* iECs in an ABO blood type-compatible rhesus monkey. a** After transplantation of human HIP\* iECs into one ABO blood type-compatible rhesus monkey, comprehensive immune analyses were performed. **b** Rhesus monkey T cell, NK cell, and macrophage killing assays as well as NK cell and macrophage ADCC and CDC assays, showed no cytotoxicity. **c** Only rhesus monkey PMNs showed strong killing activity against human HIP\* iECs. In this Figure, all rhesus macaque immune cells and serum are labeled in brown (*n* = 1 animal).

## Rhesus monkey PMNs show xeno-specific reactivity against human iECs

This observation prompted us to systematically investigate the role of macaque PMNs in human iEC xenograft rejection. Impedance cytotoxicity assays with human, mouse, and rhesus macaque PMNs as effector cells were established. PMNs are short-lived innate immune cells, and emphasis was paid to preserving their actual activation status between the isolation from rhesus macaques and the assay so that the results would reflect their function in vivo. Additionally, we aimed to simulate PMN activation in ways that are reflective of the xeno-transplant scenario (Fig. 4a). One strategy was via damage-associated molecular patterns (DAMP), those are endogenous pro-inflammatory proteins, nucleic acids, or lipids commonly released by dying graft cells after surgical trauma, ischemia, and inflammation[14–16]. Cellular stress and tissue injury thus induce perturbations of the microenvironment and activate the innate immune system[17]. DAMPs recruit PMNs from the blood to sites of tissue damage[18], where PMNs contribute to wound healing. PMNs subsequently release proinflammatory cytokines and orchestrate a diverse immune response. To activate PMNs by DAMPs in our cytotoxicity assays in vitro, we used sonicated cell debris. The second strategy to activate PMNs was via IL-2, a typical transplant-relevant pro-inflammatory cytokine. We found that rhesus PMNs rapidly killed both xenogeneic human wt and HIP\* iECs even without additional activation signals, but not allogeneic primary macaque ECs (Fig. 4b). In this allogeneic setting, macaque PMN killing of primary macaque ECs occurred only when the PMNs were activated by DAMPs or macaque IL-2. Similarly, human PMNs did not kill allogeneic human wt or HIP iECs, but PMN killing could be induced with DAMPs or human IL-2 (Fig. 4c). Also, BALB/c PMNs did not kill allogeneic B6 wt or B6 HIP iECs; killing was only observed when PMNs were stimulated. This observation hinted at the fact that the anti-human reactivity of macaque PMNs was uniquely xeno-reactive. Importantly, not all xenotransplant settings showed this xeno-responsive PMN activity. BALB/c PMNs did not kill human wt or HIP iECs—only after stimulation with DAMPs or mouse IL-2 (Fig. 4d). The human-to-mouse xenotransplant setting might thus be particularly benign.

To develop a strategy for overcoming macaque PMN killing, we tested several inhibitory PMN agents in vitro that affect PMNs via different pathways (Fig. 4e). For all of the tested agents, human or NHP safety data had been available. The five drugs CpG-52364[19], TAK-242[20],

BAY 85-8501[21], colchicine, and apocynin[22] inhibit PMN toll-like receptor (TLR)7/8/9, TLR4, neutrophil elastase, microtubule formation, and reactive oxygen species production, respectively. For each drug, a safe dose for monotherapy was anticipated from the existing literature. Unfortunately, none of the agents at the indicated concentration had a meaningful impact on rhesus PMN killing of human wt and HIP\* iECs (Fig. 4f). However, when all five drugs were combined, they effectively prevented the killing of xenogeneic human iEC target cells, even in the presence of macaque IL-2 (Fig. 4g). We then assessed the effect of this drug combination on rhesus PBMCs. In vitro priming was performed by incubating the PBMCs with the xenogeneic human wt or HIP\* iECs or allogeneic primary rhesus ECs for 14 days. CD8 + T cells were then sorted and used in cytotoxicity assays. We observed an adaptive cytotoxicity response against human wt iECs and primary rhesus ECs but not human HIP\* iECs (Fig. 4h). The drug combination did not prevent rejection of allogeneic or xenogeneic wt cells and thus did not show effective immunosuppressive activity on adaptive immune cells. On the other hand, tacrolimus, which prevents IL-2 release from T cells, and abatacept, which inhibits IL-2 receptor binding, did not show any direct effects on PMN cytotoxicity. Even the combination of both drugs did not prevent macaque IL-2-activated rhesus PMNs from killing human wt and HIP\* iECs or primary rhesus ECs (Fig. 4i). Together these results show that the 5-drug combination affected only PMN killing, but not PBMCs, and tacrolimus and abatacept only suppressed PBMCs without affecting PMN killing.

## PMN inhibition allows human HIP\* iEC xenograft survival in cynomolgus monkeys

Since we could not source any more rhesus monkeys compatible with blood type A grafts at several more primate centers, we switched to cynomolgus monkeys, which are predominantly of blood group A. Of eight cynomolgus monkeys, four received human wt iECs and four received human HIP\* iECs, but all animals were treated with the same regimen. The monkeys received a peri-transplant steroid taper and immunosuppression with abatacept and tacrolimus to prevent IL-2-activation of PMNs. They were also treated with the PMN inhibitors CpG-52364, TAK-242, BAY 85-8501, colchicine, and apocynin from 5 days before to 7 days after the cell transplantation (Figs. 5 and 6a). A total of $2 \times 10^8$ human wt or HIP\* iECs were injected into the quadriceps muscles, which are well-perfused and assumed to provide an

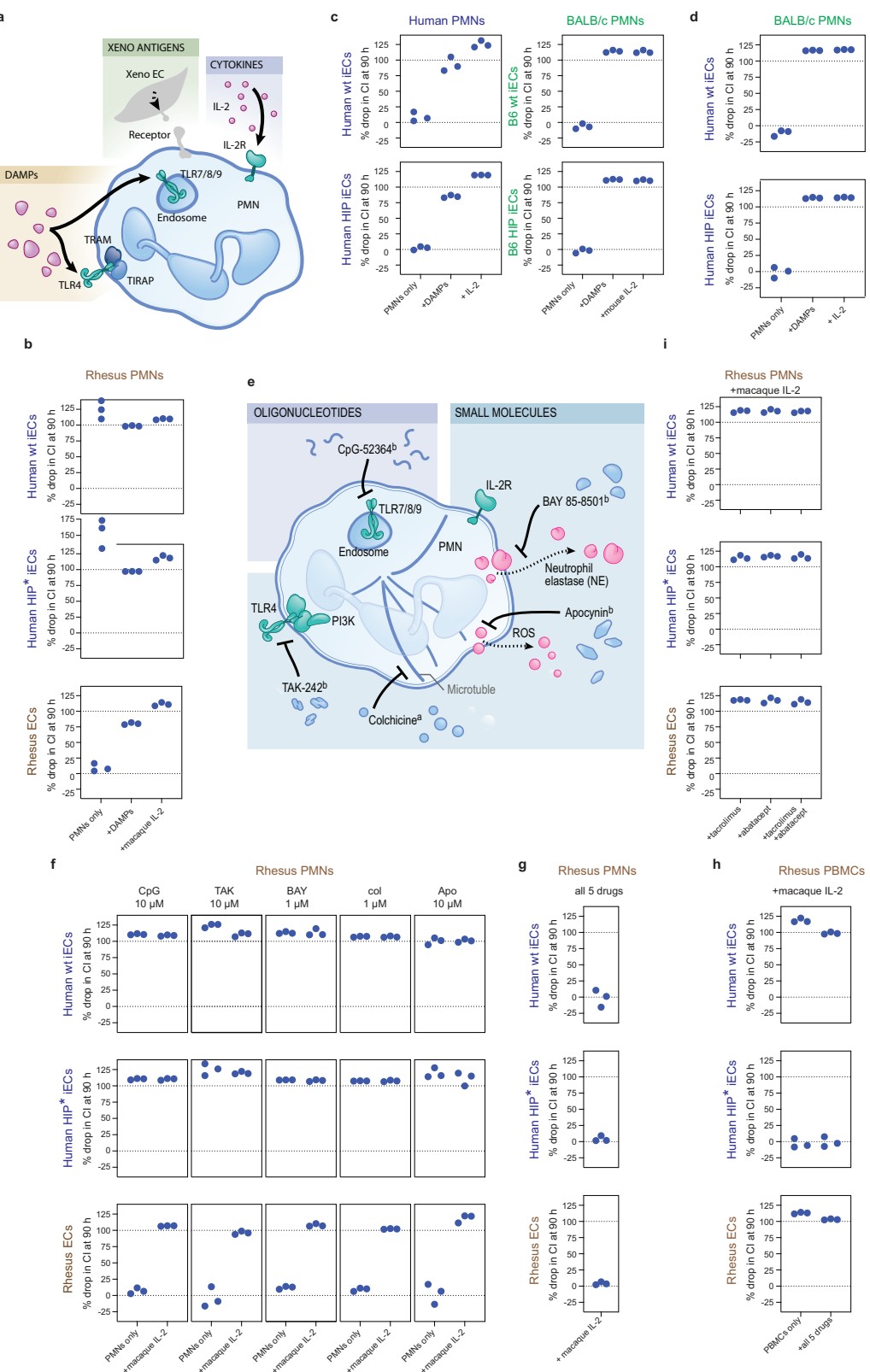

environment supportive of engraftment. The injection sites were recovered after 10 days in one monkey per group to assess early local inflammation. Histology of the HIP* iEC injection site showed columns of transplanted monomorphic CD31+ epithelioid cells with central nuclei, which were located in interfascicular spaces extrinsic to the muscle (Fig. 7a). Neutrophils were sparsely scattered in the tissues without an increased concentration in or around the injected cell

clusters (Fig. 7b). No mononuclear infiltration was seen. In contrast, no transplanted CD31+ cells could be found in the wt iEC injection site. In both recipients, areas of early teratoma growth could be identified (Fig. 7c, d). This suggests the differentiation of iPSCs into iECs was incomplete with a remaining fraction of pluripotent cells. The lower level of immunogenicity of pluripotent stem cells might have been permissive for teratoma growth in both groups[23]. Differences in the

**Fig. 4 | Mechanisms of activation and inhibition of rhesus monkey PMNs. a** We hypothesized that rhesus monkey PMNs get activated from DAMPs interacting with TLRs, from IL-2 binding to IL-2 receptors, and through unspecified xenoantigens. For these in vitro tests, sonicated target cells were added to simulate DAMP release, or IL-2 was added. **b** Rhesus monkey PMNs killed xenogeneic human wt or HIP* iECs even without prior stimulation. In contrast, they killed allogeneic primary rhesus monkey ECs only after stimulation. **c** Similarly, in human and mouse allogeneic settings, unstimulated PMNs did not kill iEC target cells. Allogeneic human and mouse PMNs exerted cytotoxicity only in the presence of DAMPs or species-specific IL-2. **d** BALB/c PMNs did not kill xenogeneic human wt or HIP* iECs in the absence of additional stimulation. Human iEC target cells were only killed with DAMPs or mouse IL-2. **e** Five drugs inhibiting different PMN activation pathways were identified. **f** The inhibitory potential of each of those drugs on rhesus PMN killing of human wt or HIP* iECs or primary rhesus monkey ECs was tested. None of the drugs showed any impact on PMN cytotoxicity in the presence or absence of macaque IL-2. **g** When all five drugs were combined, rhesus PMN killing of human wt or HIP* iECs and primary rhesus monkey ECs was suppressed, even in the presence of macaque IL-2. **h** The immunosuppressive potential of the 5-drug combination on rhesus PBMC killing of human wt or HIP* iECs or primary rhesus monkey ECs was tested in the presence of macaque IL-2. PBMCs killed allogeneic human wt iECs and xenogeneic primary rhesus monkey ECs. The five-drug combination had no effect on this PBMC response. **i** We next tested whether tacrolimus or abatacept, which inhibit IL-2 release and IL-2 receptor binding, respectively, had a direct inhibitory activity on rhesus PMNs. Macaque IL2-activated rhesus PMN cytotoxicity was not inhibited by either tacrolimus or abatacept, or a combination thereof. In this figure, all human cells are labeled in blue, mouse cells in green, and rhesus macaque cells in brown. Three different effector cell donors have been used for all graphs and individual data are shown.

immune response against iPSCs and their derivatives have to be considered in this context. All other animals completed the 28-day study period. Our immune assays (Figs. 5b–h and 6b–h) revealed a complete inhibition of cynomolgus PMNs for at least 14 days with full recovery of PMN activity at 28 days (Figs. 5e and 6e). In the HIP* iEC group, we did not detect any systemic immune activity against HIP* grafts until the PMNs recovered at the end of the study. The HIP* iEC grafts showed unrestricted survival in vivo with stable BLI signals until there was a subtle decline at day 28 (Fig. 5i). The decline of the BLI signals at day 28 correlated with the reactivated PMN activity. In the wt iEC group, the immunosuppression expectedly also inhibited T cell killing for the first week after transplantation, but the T cells recovered quickly after discontinuation (Fig. 6b). We additionally observed strong NK cell and macrophage ADCC as well as CDC killing activity against wt iECs throughout the study (Fig. 6f–h). This supports our earlier notion that macaques have pre-formed antibodies against human HLA[1]. Expectedly, we saw rapid rejection of all wt iEC grafts (Fig. 6i). Overall, these results indicated that the inhibition of PMN cytotoxicity improved the survival of human HIP* xenografts in this stringent in vivo model. However, the effective drug combination was considered too toxic for translational purposes and a drug-independent engineering approach was developed instead.

### Expression of PMN inhibitory ligands on pig wt and HIP* ECs protect from PMN killing in vitro

CD200 and CD99 are inhibitory ligands for the receptors CD200R and PILRα on immune cells including PMNs (Fig. 8a). We initially expressed CD99 and CD200 on rhesus wt ECs (rhesus wt.99.200 ECs, Supplementary Fig. 1a) for the testing in an allogeneic setting with rhesus PMNs. Indeed, this engineering effectively prevented the killing of wt.99.200 ECs by allogeneic rhesus PMNs activated with DAMPs or macaque IL-2 (Fig. 8b). In a next step, this engineering strategy was tested in a translationally highly relevant xenotransplant model. Clinical xenotransplantation of hearts and kidneys from genetically engineered pigs into patients not eligible for allotransplantation have recently been performed. We thus aimed to assess whether human PMNs would exert xeno-reactive cytotoxicity against pig wt ECs, and if so, whether CD99 and CD200 expression could inhibit that (Fig. 8c and Supplementary Fig. 1b–d). Using PMNs from 5 healthy human subjects, we could see cytotoxicity against pig wt ECs even without PMN stimulation (Fig. 8d). Primed PBMCs from 5 healthy subjects expectedly killed the pig wt ECs in what would be considered adaptive xenorejection. The engineered pig wt.99.200 ECs were successfully protected from activated human PMNs but were still rejected by primed human PBMCs (Fig. 8e). To fully overcome xenorejection, pig HIP*.99.200 ECs were engineered (Fig. 8c). Indeed, pig HIP*.99.200 ECs were fully protected from activated human PMNs as well as from primed human PBMCs in vitro (Fig. 8f).

## Discussion

Certain cell types like pancreatic islet cells[24], dopaminergic neurons[25], and cardiomyocytes[26] have been described not to express ABO blood group antigens, while some ECs do and others don't[27]. However, clinical islet transplantation is usually performed ABO-compatible[28], some clinical trials with fetal brain neural stem cells performed ABO matching[29], while trials for cardiomyocyte transplantation did not require ABO compatibility[30,31]. Thus, ABO matching has not yet been widely accepted for the transplantation of cell therapeutics although it is standard for solid organ transplantation. After the pioneering of intentional ABO-incompatible heart transplantation in infants, who did not yet produce major blood-group antibodies[32], this strategy has been adopted by many centers as standard practice and achieves similar outcomes to ABO-matched transplant[33]. In contrast, an ISHLT registry analysis of 94 accidental ABO-incompatible heart transplants in adults revealed elevated early graft failure and mortality[34]. For kidney transplantation, ABO-incompatible transplantation became possible only after the implementation of antigen-specific immunadsorption and rituximab[35,36]. These data support the importance of blood group antigens in transplantation. Macaques can have the blood groups A, B, and AB, but their frequency distributions vary significantly between rhesus and cynomolgus monkeys[37]. In this study, we show that EC xenografts need to be transplanted according to blood group compatibility.

The set of comprehensive immune assays we have developed to assess all major immune cell populations, including antibodies and serum components, correlated very well with the observed survival outcomes of cell transplant experiments in NHPs. Mechanistically, our data support the notion that PMNs contribute to the xenospecific killing of transplanted human iECs in cynomolgus monkeys. We observed that all recipient macaque PBMC populations and all serum-based killing mechanisms remained unresponsive against the HIP* iECs. The PMN population stood out as the only tested cell population that could detect human xenografts and exert cytotoxicity. In humans, neutrophils are an abundant cell population accounting for 50–70% of all circulating leukocytes[38]. They have a short average circulatory lifespan of 5.4 days[39], which increases upon activation in tissue[40]. Neutrophils recognize a variety of pathogen-associated molecular patterns (PAMP) and self-derived DAMPs through pattern recognition receptors (PRR)[41]. The response of macaque PMNs against infection is rapid and stronger than that of human PMNs[42]. PRRs include TLRs, and although macaque PMNs show a surface receptor profile somewhat different from that of human PMNs[43], their TLR expression closely mimics those of human cells[44–46] and allows the use of human PMN inhibitors. Signals initiated by DAMPs and PAMPs are transduced via similar pathways, activating innate immune inflammatory responses. Sialic acid-binding Ig-like lectins (Siglec) are inhibitory receptors that recognize sialic acids as self-associated molecular patterns (SAMP)[47] and suppress neutrophil activation[48]. Although SAMPs are not yet defined, glycans such as sialic acids and glycosaminoglycans are currently the most likely candidates[47]. Glycans might have been used to

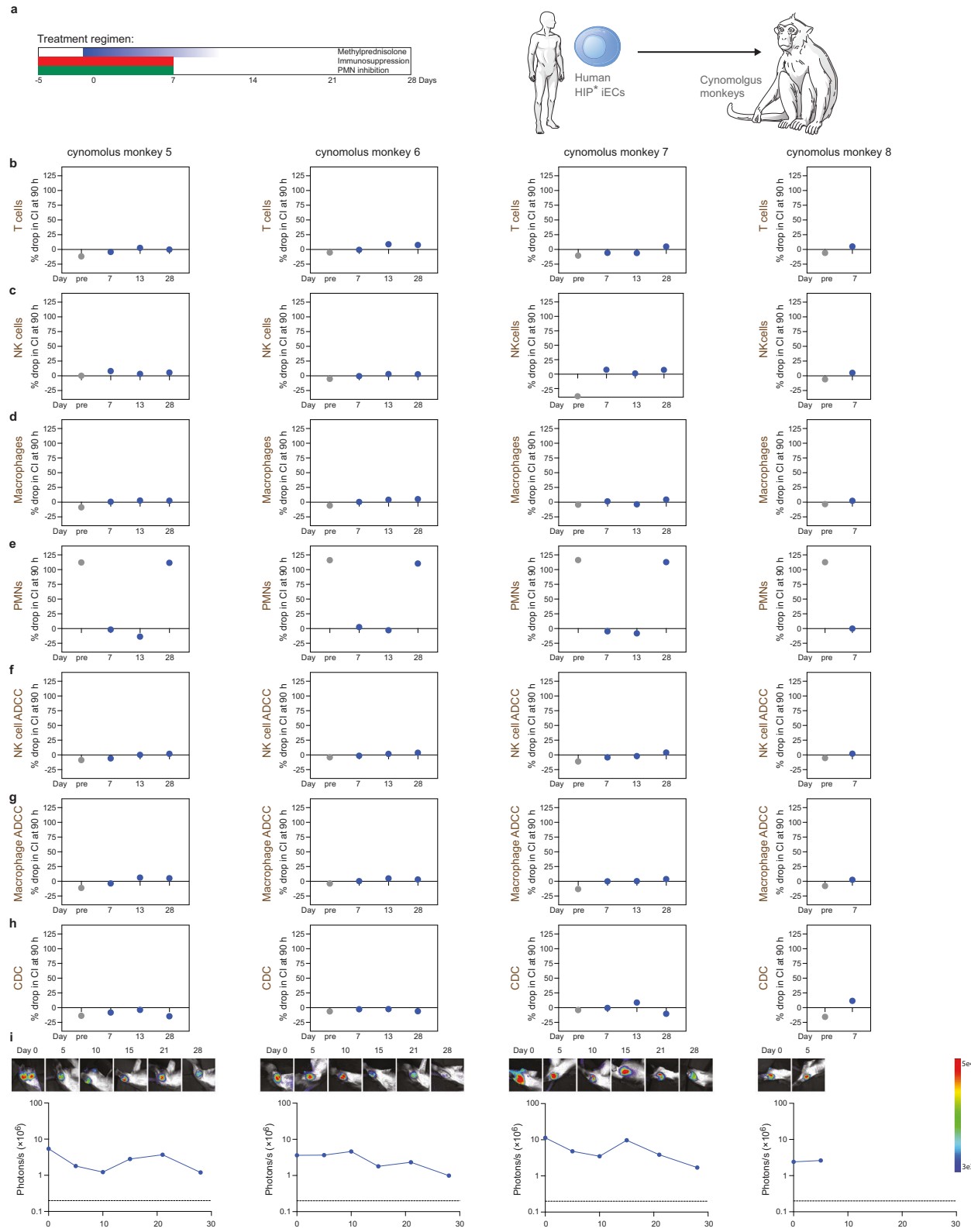

**Fig. 5 | Intramuscular transplants of human HIP* iECs in cynomolgus monkeys.**
**a** Human HIP* iECs were transplanted into xenogeneic cynomolgus monkeys. The treatment regimen for the cynomolgus monkeys included steroid taper, peri-transplant immunosuppression, and PMN inhibitors. Immunosuppression and PMN inhibitors were stopped after day 7. In vitro cynomolgus monkey T cell (**b**), NK cell (**c**), and macrophage killing assays (**d**) with human HIP* iECs showed no cyto-toxicity. **e** In vitro cynomolgus monkey PMN killing assays with human HIP* iECs showed pre-transplant cytotoxicity, which was then suppressed on days 7 and 13,

and was re-activated on day 28. In vitro NK cell ADCC (**f**) and macrophage ADCC assays (**g**) with human HIP* iECs showed no cytotoxicity. **h** In vitro CDC assays with human HIP* iECs also showed no cytotoxicity. **i** BLI imaging of the intramuscular iEC implants into the quadriceps of the cynomolgus monkeys. BLI images for the six time points are shown. The BLI signals were stable for 21 days with downward trend thereafter. In this figure, all cynomolgus macaque cells and serum are in brown (n = 4 animals).

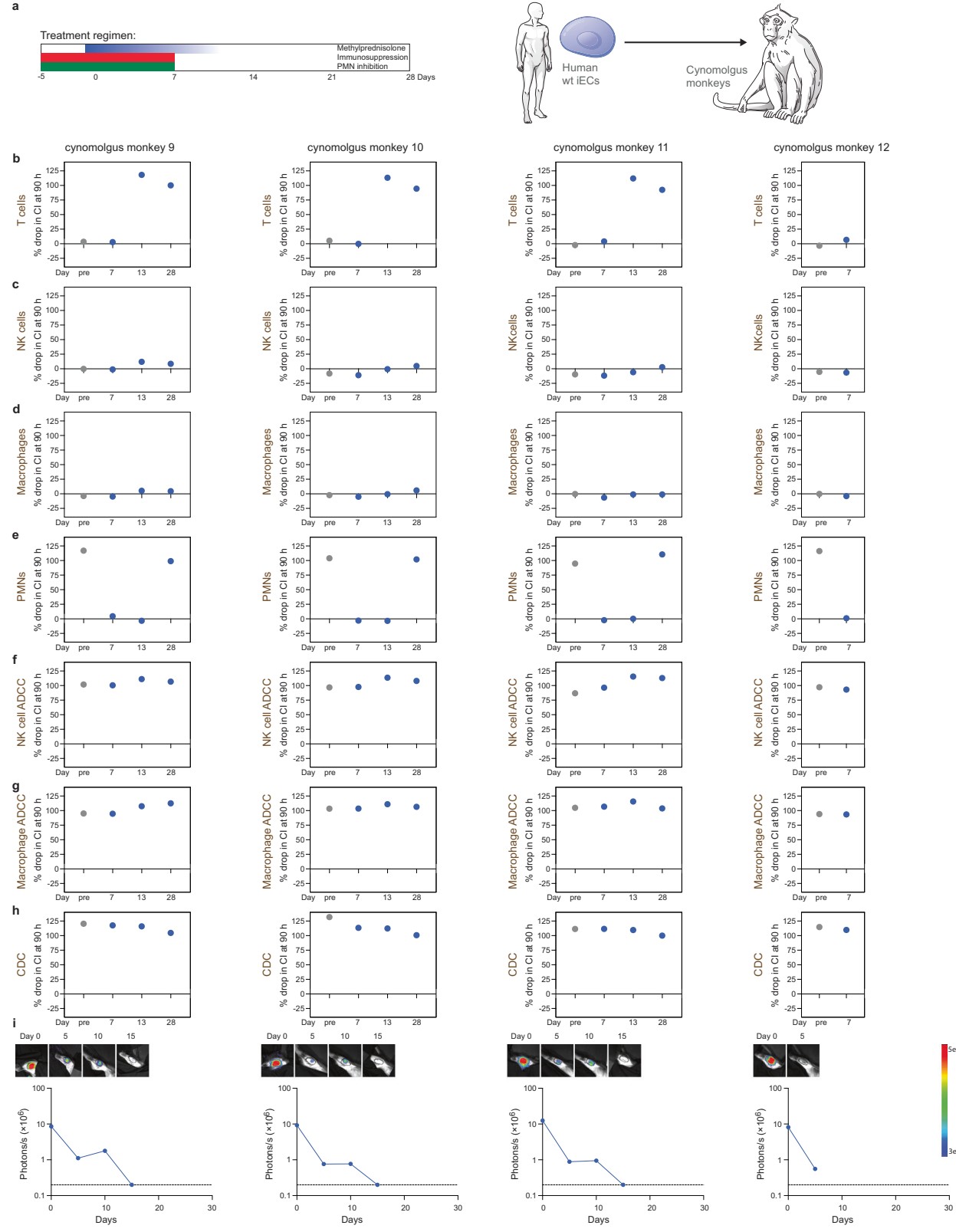

sense the presence of non-self since the beginning of multicellular life. Different cell types may be more or less recognizable by xenogeneic PMNs, depending on their specific glycome[10]. The fact that macaque PMNs similarly killed human wt and HIP* iECs in vitro shows that HLA and peptide presentation is not involved in this process.

Neutrophils have been recognized as key innate mediators in transplantation as they contribute to ischemia-reperfusion injury,

enhance adaptive alloimmunity, and impact graft survival[49]. Neutrophils are primary responders that are recruited from the circulation into the interstitium as early as 30 min after reperfusion with a peak after 24 h[50–53]. Blocking neutrophil adhesion receptors on endothelium was shown to have a protective effect in renal ischemia-reperfusion injury[54–56]. Neutrophils can cause tissue injury by plugging the microvasculature,[57] degranulating proteases,

**Fig. 6 | Intramuscular transplants of human wt iECs in cynomolgus monkeys.**
**a** Human wt iECs were transplanted into xenogeneic cynomolgus monkeys. The treatment regimen for the cynomolgus monkeys included steroid taper, peri-transplant immunosuppression, and PMN inhibitors. Immunosuppression and PMN inhibitors were stopped after day 7. **b** In vitro cynomolgus monkey T cell killing assays with human wt iECs showed a lack of pre-transplant reactivity, a suppressed T cell response after 7 days, and re-activated T cell cytotoxicity thereafter. In vitro cynomolgus monkey NK cell (**c**) and macrophage killing assays (**d**) with human wt iECs showed no reactivity. **e** In vitro cynomolgus monkey PMN killing assays with human wt iECs showed pre-transplant cytotoxicity, which was then suppressed on days 7 and 13, and was re-activated on day 28. In vitro NK cell ADCC (**f**) and macrophage ADCC assays (**g**) with human wt iECs showed cytotoxicity with pre-transplant serum, which persisted throughout the study. **h** In vitro CDC assays with human wt iECs also showed cytotoxicity with pre-transplant serum, which persisted throughout the study. **i** BLI imaging of the intramuscular wt iEC implants into the quadriceps of the cynomolgus monkeys. BLI images for the six time points are shown. The BLI signals faded over time and were all lost after 15 days. In this figure, all cynomolgus macaque cells and serum are in brown ($n = 4$ animals).

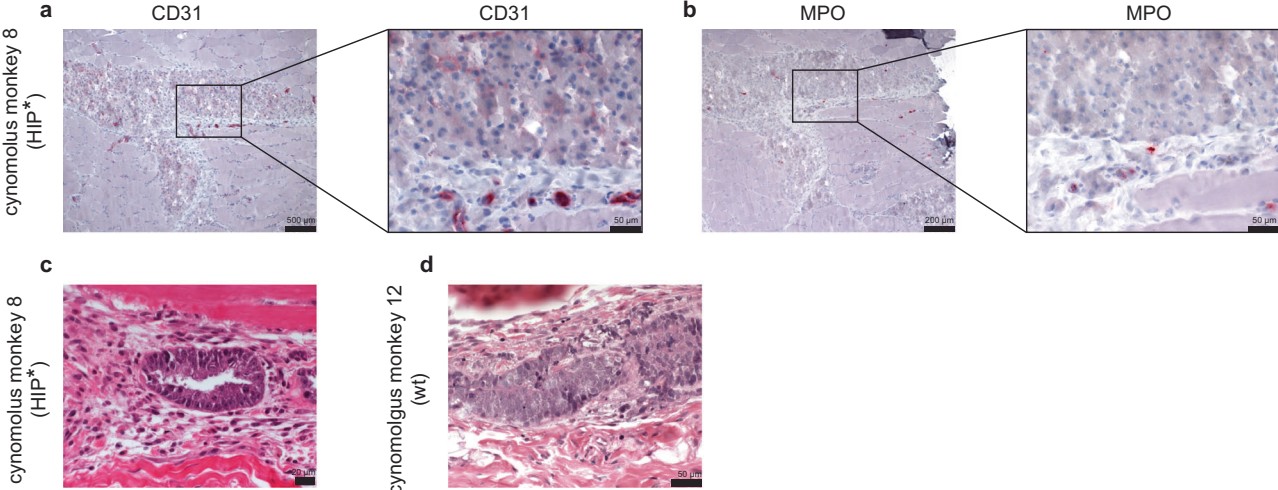

**Fig. 7 | Histology of human wt or HIP* iEC implant sites after 10 days.** The intramuscular HIP* iEC injection site was cut and stained for CD31 (**a**) and myeloperoxidase (MPO; **b**). Hematoxylin and eosin stainings showed teratomas that developed within the sites of HIP* iEC (**c**) and wt iEC injections (**d**).

myeloperoxidases and reactive oxygen radicals[58], and by forming neutrophil extracellular traps[59,60]. Neutrophil infiltration was shown to be the most important predictor of delayed graft function, and interventions reducing neutrophil infiltration could alleviate renal ischemia-reperfusion injury[61–63] and lower the incidence of delayed graft function[64]. In addition, neutrophils can initiate and enhance acute cellular allorejection[65,66] and neutrophil depletion was shown to markedly attenuate acute allograft rejection and facilitate long-term graft survival[67]. Our results reveal that PMN activation by xenoantigens further enhances their cytotoxicity and contributes to xenograft rejection.

The depletion or complete inhibition of macaque PMNs is an arduous task that has been attempted with other non-myeloablative regimens[68] or total body irradiation[69], but was not only associated with severe toxicity but also severe generalized immunosuppression. The five-drug combination used herein was more PMN-specific and less PBMC suppressive and allowed for more independent assessment of PMN function. We could show that PMN inhibition facilitated HIP* xenograft survival in cynomolgus monkeys. We assume that xeno-recognition of the HIP* iECs by PMNs had triggered the observed delayed rejection in vivo, which coincided with the recovery of the PMN killing capacity after the discontinuation of the drug regimen.

Clinical xenotransplantation is still at the pioneering stage, and our understanding of the complex immunology incomplete. Histologic findings of a recently failed genetically modified pig-to-human cardiac xenograft were not consistent with typical allorejection[70], but PMNs were not assessed. We herein show that human PMNs exert xeno-reactive cytotoxicity against pig ECs that can be overcome through the overexpression of CD99 and CD200. Pig HIP*.99.200 cells are completely protected from all human adaptive, innate, and PMN cytotoxicity and should be further assessed for clinical xenotransplantation.

## Limitations of study
We used group sizes not exceeding four NHPs throughout the study to minimize ethical concerns and we obtained early histology from only one animal per group to still allow for longitudinal graft monitoring in the remaining animals. Generalization of observations thus need to be done carefully. While our in vitro assays for PBMC subpopulations have directly correlated with the observed outcome of transplant experiments in multiple allotransplantation studies in mice[2,3,5,71] and NHPs[1], it remains unclear how well the PMN assay results translate to longer-term in vivo outcomes. We have focused on ECs given their crucial role in solid organ transplantation. It remains unclear whether PMN killing is relevant for other cell types.

## Methods
### Primary cells
Rhesus primary ECs were purchased from ATCC (CRL-1780, Cat. No. RF/6 A, ATCC, Manassas, VA) and cultured in Eagle's Minimum Essential Medium (ATCC) containing 10% FCS hi and 1% pen/strep (both Gibco, Waltham, MA). Media changed was performed twice per week. Rhesus primary PBMCs were purchased from HumanCells Biosciences (Cat. No. M5-011, HumanCells, Milpitas, CA). Pig primary ECs were purchased from Cell Biologics (Cat. No. P-6065, Cell Biologics, Chicago, IL) and cultured in complete EC medium (Cell Biologics) on gelatin coated T75 flasks. Media was changed every 2 days and TrypLE (ThermoFisher, Waltham, MA) was used for cell passaging with a ratio of 1:4 every 5 days.

### Gene editing of human, mouse, and pig HIP cells
Human wt, and $B2M^{-/-} CIITA^{-/-}$ CD47 tg (human HIP) iPSCs as well as C57BL/6 (B6) wt, and $B2m^{-/-} Ciita^{-/-}$ Cd47 tg (mouse HIP) iPSCs were generated as described[2]. The generation of human iPSCs with $B2M^{-/-} CIITA^{-/-}$ and overexpressing rhesus CD47 (human HIP*) was reported earlier[4]. The differentiation of iPSC-derived endothelial cells (iEC) was described in detail previously[2], and cultures were >95% positive for

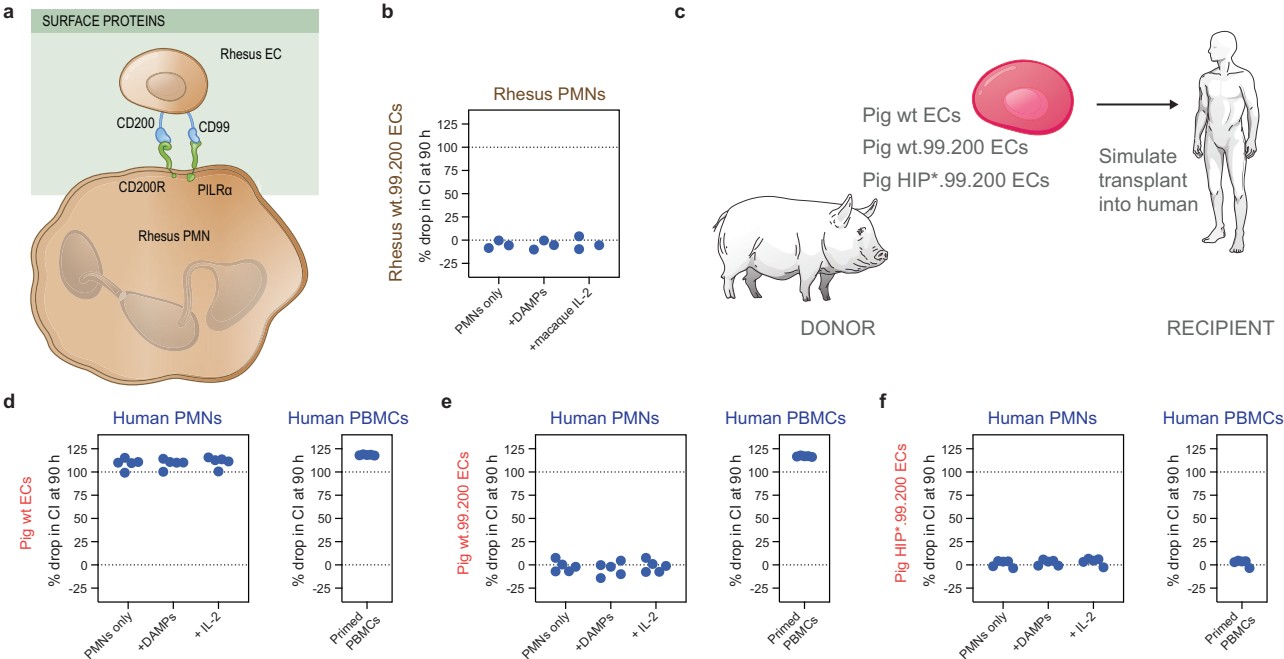

**Fig. 8 | CD99 and CD200 overexpression protect pig wt and HIP* ECs from human PMN killing. a** Human CD99 and CD200 were overexpressed on rhesus ECs for rhesus PMN killing assays. **b** Rhesus wt.99.200 ECs were protected from activated allogeneic rhesus PMNs (three rhesus PMN donors; all individual data are shown). **c** Unedited pig wt ECs and those overexpressing CD99 and CD200 (pig wt.99.200 ECs) as well as pig ECs depleted of SLA class I and II and overexpressing human CD47, CD99, and CD200 (HIP*.99.200) were incubated with human immune cells. **d** Pig wt ECs were killed already by unstimulated human PMNs and were rejected by primed human PBMCs (five human immune cell donors; all individual data are shown). **e** Pig wt.99.200 ECs were protected from human PMN killing even after stimulation but were still rejected by human PBMCs (five human immune cell donors; all individual data are shown). **f** Finally, pig HIP*.99.200 ECs were fully protected from both human PMNs and PBMCs (five human immune cell donors; all individual data are shown).

VE-Cadherin and >90% positive for CD31. Supplementary Fig. 2 shows representative flow cytometry plots. Pig HIP* ECs were similarly engineered through *B2m* and *Ciita* knockout and human CD47 overexpression.

## Transduction of CD99 and CD200
For CD99 and CD200 transduction, 100,000 pig wt or HIP* ECs or rhesus wt ECs were plated in one 6-well and incubated overnight at 37 °C at 5% $CO_2$. The next day, the media was changed, and CD99 and CD200 lentiviral particles driven by the CMV promotor (Origene, Rockville, MD) were added to 2 ml media with a MOI of 20. Complete media change was performed after 48 h. Cells were flow-sorted for the CD99 and CD200 double-positive population (BD FACSAria, BD Biosciences, San Jose, CA).

## Flow cytometry
To assess SLA class I expression, pig ECs were incubated with a AF647-conjugated anti-SLA-I antibody (clone JM1E3, Cat. No. MCA2261A647, Bio-Rad, Hercules, CA) or AF647-conjugated IgG1 isotype-matched control antibody (Cat. No. MCA928A647, Bio-Rad). To assess SLA-II expression, cells were incubated with a PE-conjugated anti-SLA-II DR antibody (clone 2E9/13, Cat. No. MCA2314GA, Bio-Rad) or PE-conjugated IgG2b isotype-matched control antibody (Cat. No. MCA691PE, Bio-Rad). For the detection of rhesus MHC-I and MHC-II on rhesus ECs, an APC-conjugated anti-HLA-A, B, C antibody (clone G46_2.6, Cat. No. 555555, BD Biosciences) or APC-conjugated IgG1 isotype-matched control antibody (clone MOPC-21, Cat. No. 554681, BD Biosciences) and an AF647-conjugated anti-HLA-DR, DP, DQ antibody (clone Tu39, Cat. No. 563591, BD Biosciences) or AF647-conjugated IgG2a isotype-matched control antibody (clone G155–178, Cat. No. 565357, BD Biosciences) were used, respectively. To assess CD47 expression, a FITC-conjugated anti-CD47 antibody (clone CC2C6, Cat. No. 323106, Biolegend, San Diego, CA) or FITC-conjugated IgG1 isotype-matched control antibody (clone MOPC-

21, Cat. No. 400107, Biolegend) was used. This antibody cross-reacts with rhesus macaque CD47, but not pig CD47. For the detection of CD99, an APC-conjugated anti-CD99 antibody (clone hec2, Cat. No. 398203, Biolegend) or APC-conjugated IgG1k isotype-matched control antibody (clone MOPC-21, Cat. No. 400119, Biolegend) was used. An APC-conjugated anti-human CD200 antibody (clone A18042B, Cat. No. 399807, Biolegend) or APC-conjugated IgG1k isotype-matched control antibody (clone MOPC-21, Cat. No. 400119, Biolegend) was used for the detection of CD200. Representative histograms are shown. The EC phenotype of human HIP* iECs was assessed using a FITC-conjugated anti-VE-Cadherin antibody (CD144) (clone QA20A44, Cat. No. 376509, Biolegend) or FITC-conjugated IgG1k isotype-matched control antibody (clone MOPC-21, Cat. No. 400107, Biolegend) and an APC-conjugated anti-CD31 antibody (clone WM59, Cat. No. 303115, Biolegend) or APC-conjugated IgG1k isotype-matched control antibody (clone MOPC-21, Cat. No. 400119, Biolegend).

## Animals
One female blood type AB rhesus monkey (*Macaca mulatta*, 2 years of age) was identified at the Oregon National Primate Research Center that was available for our study. All procedures conformed to the requirements of the Animal Welfare Act with protocols approved by the Institutional Animal Care and Use Committee at the Oregon Health and Science University. Three subcutaneous injections of $1 \times 10^7$ HIP* iECs in 500 µl 1:3 diluted Matrigel were made. The animal received no medication. Blood samples were collected pre-transplant and after 7 days, and PBMCs and PMNs were isolated.

Eight blood type A cynomolgus monkeys (*Macaca fascicularis*, 1–2 years of age, ~1–2 kg) were selected at the Alpha Genesis Primate Research Center. Five were female and three were male. All rhesus macaque experiments were approved by the Alpha Genesis Institutional Animal Care and Use Committee and regulated by the US Department of Agriculture. A total of $2 \times 10^8$ HIP* or wt iECs were

injected into the quadriceps muscle using a Hamilton pipette. Animals received a steroid taper with intramuscular methylprednisolone (30 mg/kg on day−1, then 10 mg/kg, 6 mg/kg, 3 mg/kg, and 1.5 mg/kg for 3 days each). The following PMN inhibitors were used: BAY 85-8501 0.07 mg/kg orally QD, TAK-242 5 mg/kg intramuscularly QD, colchicine 0.08 mg/kg orally QD, apocynin 100 mg/kg orally TID, and CpG-52364 1 mg/kg orally once per week. All animals received abatacept 12.5 mg/kg subcutaneously once on day−1, and tacrolimus 0.1 mg/kg intramuscularly QD). PMN inhibitors and immunosuppression was discontinued on day 8. Scheduled BLI imaging was performed, and blood samples were collected pre-transplant and after 7, 13, and 28 days, and PBMCs and PMNs were isolated.

## Monkey BLI imaging
D-luciferin firefly potassium salt (375 mg/kg, Biosynth, Staad, Switzerland) was dissolved in sterile PBS (pH 7.4, Gibco, Invitrogen) and was injected intravenously into anesthetized monkeys. Animals were imaged using the Lago (Spectral Instruments Imaging, Tuscon, AZ) 12 min after the d-luciferin application for an exposure of 5 min. Region of interest bioluminescence was quantified in units of maximum p/s/cm2/sr. The maximum signal from a region of interest was measured using Aura 3.2 software (Spectral Instruments Imaging).

## Rhesus NK cell isolation
Rhesus PBMCs were isolated from fresh blood by Ficoll separation. Cells were sorted on the FACSAria Fusion using a FITC-conjugated anti-CD8 antibody (clone LT8, Cat. No. ab28010, Abcam, 1:5, Cambridge, Great Britain) and a PE-conjugated anti-NKG2A antibody (clone REA110, Cat. No. 130-114-092, Miltenyi, 1:50, Bergisch Gladbach, Germany) to select a CD8+ NKG2A+ NK cell population as previously reported[72].

## Rhesus T cell isolation
Rhesus PBMCs were isolated from fresh blood by Ficoll separation. Cells were sorted on the FACSAria Fusion using an APC-conjugated anti-CD3 antibody (clone 10D12, Cat. No. 130-123-790, Miltenyi, 1:50). T-cells were cultured in OpTmizer media (Gibco), containing 100 IU/mL rhesus IL-2 (Cat. No. MBS1376561, MyBiosource, San Diego, CA).

## Isolation of human, rhesus or mouse PMNs
Human, rhesus, or mouse PMNs were isolated from fresh blood. For in vitro rhesus macaque PMN experiments, several blood type ABO rhesus macaques were used as blood donors. In a 15 ml conical tube, Leuko-Spin and Lympho-Spin were mixed in a ratio of 1:1 (both PluriSelect, Leipzig, Germany). Fresh blood was layered on top of the Leuko-Lympho-Spin mix, followed by centrifugation for 30 min at 1000 × g with brake off. After the density centrifugation, PMNs can be found on the lower layer. The DURAClone IM Granulocyte kit (Cat. No. B88651, Beckman Coulter, Brea, CA) was used to assess the fractions of granulocyte subpopulations according to the manufacturer's instructions (Supplementary Fig. 3).

## Macrophage differentiation from rhesus PBMCs
PBMCs were isolated by Ficoll separation from fresh blood and were resuspended in RPMI-1640 with 10% heat-inactivated fetal calf sera (FCS hi) and 1% pen/strep (all Gibco). Cells were plated in 24-well plates at a concentration of $1 \times 10^6$ cells per ml and medium was changed every second day until day 6. Macrophages were stimulated from day 6 with 10 ng/ml rhesus M-CSF (cat.no. orb1955050, Biorbyt, Cambridge, Great Britain) and 1 µg/ml rhesus IL-2 (Cat. No. MBS1376561, MyBiosource) for 24 h before the cells were used for assays.

## Ex vivo human or macaque T cell priming
Human or rhesus PBMCs were obtained after Ficoll separation of human or rhesus monkey blood draws. For priming, $1 \times 10^6$ rhesus

PBMCs were co-cultured with $5 \times 10^5$ human wt iECs in gelatin-coated flasks. Alternatively, $1 \times 10^6$ human PBMCs were co-cultured with $5 \times 10^5$ pig wt iECs, pig HIP* iECs. The media, which consisted of a 1:1 mixture of EC medium and PBMC medium, was changed every 3 days. After 14 days, the cells in suspension were harvested and sorted using an APC-conjugated mouse anti-human CD3 antibody (clone SP34-2, Cat. No. 557597, BD Biosciences, concentration 0.01 mg/ml) together with the APC-conjugated IgG1κ isotype-matched control antibody (clone MOPC-21, Cat. No. 550854, BD Biosciences, concentration 0.01 mg/ml) and a BV421-conjugated anti-human CD8 antibody (clone SK1, Cat. No. 344748, BioLegend, concentration 0.005 mg/ml) together with the BV421-conjugated IgG1κ isotype-matched control antibody (clone MOPC-21, Cat. No. 400157, BioLegend, concentration 0.005 mg/ml). Both antibodies were cross-reactive with macaque T cells. The CD3+CD8+ cells were sorted using a FACSAria flow cytometer (BD Biosciences) and used for real-time XCelligence killing assays.

## Cytotoxicity assays on the XCelligence platform
In vitro killing assays were performed on the XCelligence SP platform and MP platform (ACEA BioSciences, San Diego, CA). Special 96-well E-plates (ACEA BioSciences) were coated with collagen (Sigma-Aldrich, St. Louis, MO), and $4 \times 10^5$ mouse wt or HIP iECs, rhesus primary ECs, or human wt, HIP, or HIP* iECs were plated in 100 µl cell-specific medium. After the CI reached 0.7, effector cells were added with an effector-to-target cell ratio of 1:1. NK cells or PMNs if indicated were stimulated with 1 µg/ml rhesus IL-2 (Cat. No. MBS1376561, MyBiosource,) or 1 µg/ml mouse IL-2 (Cat. No. 212-12-20UG, Peprotech, Cranbury, NJ). In some experiments, PMN inhibitors were added (BAY 85-8501 1 µM, TAK-242 10 µM, colchicine 1 µM, apocynin 10 µM, and CpG-52364 10 µM). For ADCC or CDC assays, serum that was complement-depleted or complement-preserved, respectively, was added. Data were standardized and analyzed with the RTCA software (ACEA BioSciences). In dot graphs, the percent drop in CI at 90 h is shown. Although target cell killing ranges between 0% and 100%, values below 0 and above 100 can occur in this calculation. Negative values reflect an increase in CI from proliferation of the plated cells or change in their cell morphology that enhances coverage of the electrodes. Values above 100 reflect increased conductivity from cellular electrolytes after complete disintegration of the target cells.

## Histology
Muscular autopsy sections of injection sites were obtained, fixed in 4% paraformaldehyde, dehydrated, and embedded in paraffin. Each block was sectioned into five-micrometer sections followed by hematoxylin and eosin staining.

## Immunohistochemistry
Paraffin sections underwent heat-induced antigen retrieval with Dako antigen retrieval solution pH9 (Cat. No. S2368, Dako, Santa Clara, CA) followed by blocking with blocking solution (Cat. No. POLAP-006, Zytomed Systems, Berlin, Germany). Primary antibodies were used as appropriate: Myeloperoxidase (MPO) (Cat. No. PA5-16672, Invitrogen), CD31 (Cat. No. ab28364, Abcam). Dako New Fuchsin Substrat-Chromogen System (Cat. No. K0698, Dako) was used for visualization. Imaging was performed using a Leica DMi8 microscope (Leica, Wetzlar, Germany).

## Quantification and statistical analysis
All individual data points are shown, and where pertinent, mean ± SD were added. GraphPad Prism 9 was used for all analyses. Animals were randomly assigned to experimental groups. The number of animals per experimental group is presented in each figure.

## Reporting summary

Further information on research design is available in the Nature Portfolio Reporting Summary linked to this article.

## Data availability

All data generated or analyzed during this study are included in this published article (and its supplementary information files). Source data are provided with this paper.

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

## Acknowledgements
The work was supported by a grant from the NIH (R01HL140236) to T.D. Medical Illustrations (Figs. 1, 3a, 4a, e, 5a, 6a and 8a, c) by Justin A. Klein, CMI/Emily Cheng, CMI for Mito Pop fall under the CC BY-NC-ND 4.0 licence.

## Author contributions
X.H., G.T., and A.G. performed the immunobiology experiments, and cell culture work. G.T., J.S., and T.J.M. performed monkey surgeries and monkey bioluminescence imaging. G.T. did the histology work and A.J.C. analyzed the histopathology. T.D. and S.S. designed the experiments, supervised the project, and wrote the manuscript. All the authors interpreted the data, read, and provided feedback on the figures and manuscript.

## Competing interests
Sana Biotechnology, Inc. has an exclusive license on the HIP cell technology. S.S. and X.H. are currently employees of Sana Biotechnology, Inc. S.S., T.D., X.H., and T.J.M. own stock in Sana Biotechnology, Inc. Data to Fig. 2 were generated at Sana Biotechnology, Inc., all other data were generated at UCSF. The University of California, San Francisco has filed patent applications that cover these inventions. Correspondence and requests for materials should be addressed to S.S. (Sonja.Schrepfer@cshs.org). The remaining authors declare no competing interests.
