## [Transparent Peer Review file · Nature Communications]

Inhibition of polymorphonuclear cells averts cytotoxicity against hypimmune cells in xenotransplantation

Corresponding Author: Professor Sonia Schrepfer

Version 0:

Reviewer comments:

Reviewer #1

(Remarks to the Author)

Hu et al report on inhibition of polymorphonuclear cells to decrease rejection of HIP cells in a NHP model. The authors used engineered (human) hypimmune endothelial cells that overexpress macaque CD47 (HIP*) in order to prevent xenogeneic response when transplanted into NHPs.

The authors show that HIP* cells induce a strong xenogeneic response mediated (solely) by PMNs. HIP* iEC xenograft survival in NHPs could be achieved by pharmacologic PMN inhibition. In a translational approach, overexpression of CD200 and CD99 on pig ECs prevented human PMNs from killing the pig cells in vitro.

This group has published several highest quality papers on this matter and the whole study seems to be of highest quality in terms of methods and conduction of experiments.

However, the paper is primarily descriptive and adds details on some questions and although all results are interesting, the real novelty is missing here.

The HIP* ECs have been described before and for most of the manuscript the authors are showing what everybody was expecting e.g. ABO experiments in Fig. 1. Figure 1 could be condensed or moved to the supplements as the most important line in the results section is maybe that tab-mediated killing is independent of HIP* editing in ECs and iECs.

Although the authors state the limited number of animals (for in vivo exp) at the end of the manuscript, they do not mention number of animals or of biological replicates, number of experiments or sex as a variable for any other assay.

In general, I think that this is interesting work but it might not overly excite the majority of readers as xenotransplantation is a niche field (despite the media hype) and even among experts the predictive value of NHP studies is controversial.

The translational aspect of bringing in a porcine-to-human angle is exciting but the red line connecting it to the rest of the manuscript is missing.

All together it looks like this manuscript was hastily prepared with data that did not fit in the already published papers or is not enough for publication on its own. Some minor points are that the Fig. legend doesn't match the Fig (Fig. 1).

Reviewer #2

(Remarks to the Author)

In this manuscript, Hu and colleagues report an elegant series of experiments demonstrating the ability of PMNs to mediate cytotoxic responses to xenogeneic cells modified to evade other immune responses. This is an important finding with implications for both the testing of engineered hypimmune cell therapies and for xenotransplantation. The experiments and results are clearly presented and follow a logical progression. Despite the importance of the observation, the authors admit that generalization of their findings is challenging with the data available for a few reasons: 1) the interaction appears to be species specific, as NHP PMNs killed the human HIP* cells, but a similar effect was not seen with BALB/c PMNs. 2) Sample sizes are limited, as they are in virtually all NHP studies. 3) While PMN-dependent killing could be inhibited pharmacologically or via engineered expression of inhibitory molecules, the underlying mechanism/pathway remains unclear.

In regards to limitation 1, the authors rightly looked to the pairing of human PMNs with porcine endothelial cells, given that pig-to-human xenotransplantation is the most clinically-relevant. I agree the 5-drug regimen demonstrated to be effective in vitro would not be clinically appropriate. However, in the cynomolgus transplant experiments, the animals received both this 5-drug regimen and conventional immunosuppression with tacrolimus and abatacept. The relative importance of these in abrogating the PMN response is unknown (the tacrolimus and abatacept were not tested in vitro). Additionally, in the clinically-relevant pig-to-human xenotransplantation scenario, multiple other genetic modifications to the source animal have been made, including human CD47 and other anti-inflammatory molecules. Whether these genetic modifications with additional pharmacologic immunosuppression could prevent PMN-mediated killing remains unknown.

In regards to limitation 2, were all the PMNs used for the in vitro experiments isolated from the single ABO-compatible Rhesus macaque? If so, this significantly limits the generalizability of the results, even though the cynomolgus experiments yielded consistent results.

A final point, these observations were borne out of the goal of establishing a NHP model in which to test HIP therapeutics. In doing so, the authors identified yet another key difference in immunobiology between humans and NHP. As increasingly sophisticated and targeted therapies are developed for use in humans, the ability to test these therapies in animal models will become increasingly difficult. In xenotransplantation, recent attention has turned to the use of human decedents as a model in which xenografts or other therapeutics could be tested in a human, at least for a short period of time.

Despite these limitations, the underlying observation remains important to multiple fields and reveals another layer of complexity in our understanding of innate immunity. I applaud the authors for their work.

Minor/Typographic Points:

- Figure 4A caption "Primar"
- Supplemental Figure 1 caption: the panels are incorrectly referenced in the caption as "A" and "B" rather than "A", "B", and "C"
- Supplemental Figure 3: the caption refers to panels A, B, C, and D while the figure is labeled H, I, J, and K.
- In the methods, a few portions are duplicated:
 - lines 512-515 and 518-521
 - lines 526-533 and 542-547

Reviewer #3

(Remarks to the Author)

This is an important study that experimentally interrogates the role of polymorphonuclear cells (PMNs) in allojection of wild type and hypimmune gene edited pluripotent stem cell (PSC)-derived cell therapies. PMNs play an established role in transplant allojection, however, they are not well-studied in the context of PSC regenerative medicine therapies. This manuscript has the potential to add important insights to the field with regard to PMNs in the context of xenotransplantation and regenerative medicine. However, there are significant weaknesses that need to be addressed prior to publication.

It is unclear why there is a need to study the non-human primate (NHP) immune responses to human therapies, and why additional NHP-specific gene edits (e.g., macaque CD47) should be edited into human PSCs. Human cell transplant studies have been conducted in NHPs, and NHP-to-NHP studies have been conducted to mimic the robust patient physiology and immune responses, with the goal of translating the findings to the clinic. Addition of NHP-specific edits into human cells is not clinically relevant, and the practical value and even basic science understanding aspect of advancing these mechanistic studies is unclear. This should be justified much more clearly.

One thing to consider is that this manuscript could be split into two separate manuscripts--one focused on human into NHP transplants and the other looking at CD200 and CD99 overexpression to overcome PMN rejection responses in the setting of pig xenotransplantation into humans. There is a tie-in with the PMNs in both studies, but the latter is more clinically relevant to pig xenotransplantation—a very intriguing application of this technology. Consider separating this into two manuscripts, where each section could be more-fully developed, otherwise the connection between the two should more clearly explained.

Assuming the manuscript will proceed as written, PMNs should be discussed in greater detail. Are neutrophils the key PMN mediator of these responses or are other granulocyte cell types involved? This should be discussed and at minimum there should be profiling (flow cytometry) of the PMNs to show the various subsets, and to see how these vary between individual monkeys (Fig 3) and humans (Fig 4). If it is not possible to get samples from the exact animals and human donors for these experiments, multiple replicates should be included from representative samples to show the readers the composition of the PMNs and how they vary between individuals.

The development of teratomas is a significant safety concern. Any potential cell therapy will need to be characterized for phenotype and purity. This information is lacking from the manuscript, and the formation of teratomas illustrates why this important information is needed. Please add methods describing the characterization done in your experiments, and data (in figure, table, or text form) illustrating the purity of the cells used in all experiments. This could be added in the methods (e.g., "for these experiments ECs are classified as CD31+ CD45- etc cells, experiments used 90% +/- 10% pure EC cells...")

An additional experiment should be added to Figure 4 to assess whether CD200 and CD99 overexpression in pig ECs has a protective effect against not just human PMNs, but human PBMCs. The receptor for CD200 is expressed not only on neutrophils but also macrophages, and some T cells and B cells, so there may be a benefit above PMNs extending to other effector immune cell types.

The abstract states, in reference to Figure 2, that “macaque PMN response was...not observed with macaque EC allografts in vitro.” However, that is not completely accurate since a response was shown in the (arguably more-clinically relevant) conditions of +DAMPs and +IL2 (both of which would be present in the inflammatory post-surgical environment encountered by various types of PSC grafts). This should be clarified.

An additional experiment should be conducted, or at least discussed, where CD200 and CD99 overexpression is added to NHP cells and put into the in vitro assay with NHP PMNs + DAMPS and + IL2. The authors' very encouraging finding with the pig cells would suggest that these additional edits could confer protection in the 100% NHP setting (i.e., edited NHP PSC-ECs and NHP PMNs). Further, the entirely NHP study would have direct clinical relevance owing to NHPs well-established use in pre-clinical pipelines. The current NHP vs human xeno focused study seems to miss an opportunity to study these promising new edits in an NHP model system that could be use to justify human clinical trials.

This manuscript is important because of the focus on PMNs, including neutrophils. However, authors state that PMNs are “not usually associated with graft rejection” yet there is a substantial body of literature describing the important role that neutrophils play in solid organ transplant rejection. The text should be modified to correct this error and to include relevant references (e.g. <https://doi.org/10.1038/s41423-023-01058-1> ; doi: 10.1111/ajt.13940 ; <https://doi.org/10.1016/j.trim.2023.101898>). Additionally, there is at least one other report of evaluation of neutrophils in the context of PSC hypimmune therapies (doi: 10.1101/2024.06.07.597791)—this should be cited as it highlights the authors' case that this is an important and growing area of research relevant to regenerative medicine.

It is unclear from the figure legend in Figure 2 why the scale for in vitro tests goes to 125% instead of 100% and why one sample in B is offscale. Please clarify.

The role of ABO mismatch/match should be discussed in more detail.

Version 1:

Reviewer comments:

Reviewer #1

(Remarks to the Author)

I have to admit that the authors tried to satisfy the reviewers and put a lot of work and effort in the revision and point to point reply. As for some concerns of the reviewers the did a great job, however, for other party of the manuscript it became even more confusing. The manuscript was rewritten to consist of 2 parts like suggested by another reviewer, which increases my concerns about the first part of the manuscript: NHP-specific edits into human cells is for human-to-NHP xenoTX studies not clinically relevant. The importance of PMN is undisputed, but showing this in the translationally relevant pig-to-human model would have been more interesting for the (small) XenoTX community.

The authors state that “We thus do believe that this publication will help promote the awareness of ABO being important for cell transplantation.” There is a huge amount of data and expertise on the topic of ABO matching for HSCT, islet cell transplants,..., the concept of necessity for matching has also been challenged in solid organ transplantation (e.g. Lory West in pediatric HTx, enzymatic conversion of blood groups of human kidneys,...). The fact that NHP centers don't do it on a regular basis does not preclude the importance of the concept of incompatibility that is, like in humans, elegantly shown by the authors in new fig.S1A.

The number of animals as well as information about the sex in the in vivo experiments are now mentioned in the Methods. The authors also added number of animals in the Figure legends. Even though this was what I requested, the fact that Fig.2 is dedicated to the results of ONE single animal is something that (at least in my personal opinion and I am really kind of sorry for that) makes the reader suspicious and does not help the credibility of the data or manuscript.

Minor point: the title does only cover for one part of the work now.

Although interesting from a basic science perspective (and without limiting the authors credibility or scientific impact and importance), I still have major concerns that this manuscript provides robust data that allow generalized conclusions about the use of HIP cells between different species and the impact for the field of xenotransplantation.

Reviewer #2

(Remarks to the Author)

I appreciate the authors thoughtful responses to all reviewers comments. I believe all of the comments and critiques have been adequately addressed. The manuscript has been strengthened and the results are more clearly presented. Thank you for the opportunity to review this work.

Reviewer #3

(Remarks to the Author)

The revised manuscript has meaningfully improved compared to the original submission. The comments to the reviewers were near satisfactory but multiple responses need further clarification. Most importantly, in multiple instances the authors state that they have revised the manuscript in light of specific reviewer suggestions but no page/line #s are given so it cannot be verified whether satisfactory changes have been made or not. For every mention of the manuscript being revised, please provide specific page and/or line numbers (see below for more details).

Please address the following follow-up questions related to specific author responses.

3.1: Please point out the specific parts of the manuscript structure that was revised.

3.2: Please point out the specific parts of the manuscript structure that was revised.

3.3: This figure is very helpful. Please update it further to add percentages in the all gates (currently, you only show percentages for the eosinophils, and in the bar graphs). The figure should show the percents of each cell type, since it is difficult to determine exactly from the bar graphs. Given that the vast majority of your PMNs are neutrophils, this is now appropriately emphasized in the manuscript Discussion. It is helpful to discuss specific cells within the PMN population, as each cell type plays a unique role and their frequencies vary considerably within a given blood sample i.e., neutrophils comprise the vast majority of PMNs.

3.4: I disagree with the authors' assessment that immunology studies should not include analysis of cell purity. Graft composition, especially with regards to the purity of the cells within, is helpful and important information in the context of immunology. Arguably, teratoma formation due to presence of undifferentiated PSCs is MORE of a concern in the context of hypoimmune cell therapies because of the potential for unchecked growth of now-hypoimmune tumors. Additionally, prior research has shown that there can be significant differences in how the immune system responds to PSCs vs PSC-differentiated cells from the same donor. See PMID: 23371903 for more information regarding immunogenicity of PSCs vs their differentiated products. Please cite this manuscript and also clarify in the methods whether the purity shown in the Supplementary Figure 4 is representative of a typical differentiation, or if that varies and it is possible that your manuscript contains impure cell populations. Highly-pure cell preparations will be a requirement for any cellular therapy (i.e., an overarching goal of this work).

3.6: Please note which lines were adjusted.

All other questions were addressed satisfactorily.

Version 2:

Reviewer comments:

Reviewer #1

(Remarks to the Author)

In the revised version the authors provided improved rationale for the studies with regard to the clinical relevance of the pig-to-human model. Moreover, the point to point reply contains a very interesting summary of the rationale and importance of ABO matching (which might also be interesting for others and could (in part?) be included into the manuscript?

Thank you for clarification on the animal numbers which in my opinion is crucial in preclinical animal studies.

In summary this version is significantly improved compared to the initial manuscript.

Reviewer #3

(Remarks to the Author)

The authors have sufficiently addressed all of my comments, with one exception.

The following portion should be updated to acknowledge that complexity of their experimental model:

page 2 line 30: "Transgenes expressed on cellular grafts must be of the recipient species to properly interact with the recipient immune system or have to be sufficiently crossreactive to be functionally competent."

The concept of modifying human cells to work in NHP potentially makes it less clinically relevant than had they used entirely human cells/human edits in the NHP system, as other preclinical studies have done. (see Zhu et al 2018 Circ Res; Du et al 2022 Nat Med). Making human cells work better in NHP than they otherwise would have doesn't really have an analogy in the human clinical setting. In fact, the edits described potentially introduce additional, new variables that would not be

encountered in an entirely human setting. Please address this directly, in the context of your overall goal of creating human therapeutics.

Dear Reviewers:

Thank you very much for your review of our manuscript **NCOMMS-24-62524** for publication in Nature Communications.

We have taken all comments made by the Editor and the Reviewers very seriously and have performed several additional studies that were suggested by the reviewers, have restructured the manuscript, revised the title, and have expanded on the discussion to reflect the reviewers' remarks.

Please find a detailed point-by-point response to the comments (*in blue italics*) below.

Reviewer #1 (Remarks to the Author):

Hu et al report on inhibition of polymorphonuclear cells to decrease rejection of HIP cells in a NHP model. the authors used engineered (human) hypoimmune endothelial cells that overexpress macaque CD47 (HIP) in order to prevent xenogeneic response when transplanted into NHPs. The authors show that HIP* cells induce a strong xenogeneic response mediated (solely) by PMNs. HIP*iEC xenograft survival in NHPs could be achieved by pharmacologic PMN inhibition. In a translational approach, overexpression of CD200 and CD99 on pig ECs prevented human PMNs from killing the pig cells in vitro.*

This group has published several highest quality papers on this matter and the whole study seems to be of highest quality in terms of methods and conduction of experiments.

However, the paper is primarily descriptive and adds details on some questions and although all results are interesting, the real novelty is missing here.

*1.1 The HIP*ECs have been described before and for most of the manuscript the authors are showing what everybody was expecting e.g. ABO experiments in Fig.1. Figure 1 could be condensed or moved to the supplements as the most important line in the results section is maybe that ab mediated killing is independent of HIP*editing in ECs and iECs.*

We have moved the ABO Figure to the Supplement as **Supplementary Fig. 1**. The reviewer is obviously very much up to date on the ABO issue. That, however, has not been the commonly accepted notion in the xenotransplant field during the time of this study. When we contacted colleagues that had published human somatic cell transplants in non-human primates, none had taken ABO into consideration. Also, none of the primate centers we have worked with had done ABO typing before we asked them to. We thus do believe that this publication will help promote the awareness of ABO being important for cell transplantation.

1.2 Although the authors state the limited number of animals (for in vivo exp) at the end of the manuscript, they do not mention number of animals or of biological replicates, number of experiments or sex as a variable for any other assay.

The number of animals in the in vivo experiments is now mentioned in the **Methods**. We have added how many were female and male. For all in vitro experiments, biological replicates and technical replicates are now spelled out in the **Figure legends**.

1.3 In general, I think that this is interesting work but it might not overly excite the majority of readers as xenotransplantation is a niche field (despite the media hype) and even among experts the predictive value of NHP studies is controversial.

The translational aspect of bringing in a porcine-to-human angle is exciting but the red line connecting it to the rest of the manuscript is missing.

Thank you very much for pointing this out. The manuscript has been restructured and more emphasis is now being placed on the engineered pig cells. We explain that for in vivo experiments, the established human-to-NHP model was used, but the story line leads to the engineering of pig cells. We have added more pig EC groups including pig HIP*.99.200 ECs, that are fully protected against all human adaptive, innate, and PMN cytotoxicity (see **Fig. 6**).

1.4 All together it looks like this manuscript was hastily prepared with data that did not fit in the already published papers or is not enough for publication on its own. Some minor points are that the Fig. legend doesn't match the Fig (Fig.1).

The Figure legends were checked, corrected, and all Figures and legends were formatted according to the journal's guidelines.

This was a long study conducted over several years and this manuscript summarizes the finding we learned with regard to ABO and PMNs. We have taken the reviewer's criticism to heart and have revised the structure of the manuscript with a clear red line now towards engineering of pig cells for xenotransplantation into humans.

Reviewer #2 (Remarks to the Author):

In this manuscript, Hu and colleagues report an elegant series of experiments demonstrating the ability of PMNs to mediate cytotoxic responses to xenogeneic cells modified to evade other immune responses. This is an important finding with implications for both the testing of engineered hypimmune cell therapies and for xenotransplantation. The experiments and results are clearly presented and follow a logical progression. Despite the importance of the observation, the authors admit that generalization of their findings is challenging with the data available for a few reasons: 1) the interaction appears to be species specific, as NHP PMNs killed the human HIP cells, but a similar effect was not seen with BALB/c PMNs. 2) Sample sizes are limited, as they are in virtually all NHP studies. 3) While PMN-dependent killing could be inhibited pharmacologically or via engineered expression of inhibitory molecules, the underlying mechanism/pathway remains unclear.*

2.1 In regards to limitation 1, the authors rightly looked to the pairing of human PMNs with porcine endothelial cells, given that pig-to-human xenotransplantation is the most clinically-relevant. I agree the 5-drug regimen demonstrated to be effective in vitro would not be clinically appropriate. However, in the cynomolgus transplant experiments, the animals received both this 5-drug regimen and conventional immunosuppression with tacrolimus and abatacept. The relative importance of these in abrogating the PMN response is unknown (the tacrolimus and abatacept were not tested in vitro).

We have performed the experiments the reviewer was suggesting and present them in **Fig. 2i**. The results confirm our assumptions that tacrolimus and abatacept do not directly inhibit PMN activity.

2.2 Additionally, in the clinically-relevant pig-to-human xenotransplantation scenario, multiple other genetic modifications to the source animal have been made, including human CD47 and other anti-inflammatory molecules. Whether these genetic modifications with additional pharmacologic immunosuppression could prevent PMN-mediated killing remains unknown.

We agree with the reviewer. It is exciting to see engineered pigs with 10 gene edits (by Revivicor) or 69 gene edits (by eGenesis) tested for xenotransplantation. Unfortunately, we do not have access to these pigs and thus cannot test their resistance against human PMN killing. This manuscript may motivate such testing and additional edits can be added if shown to be beneficial.

2.3 *In regards to limitation 2, were all the PMNs used for the in vitro experiments isolated from the single ABO-compatible Rhesus macaque? If so, this significantly limits the generalizability of the results, even though the cynomolgus experiments yielded consistent results.*

For in vitro PMN experiments with rhesus PMNs, several blood type ABO rhesus macaques were used. All experiments shown in Fig. 2 were done with 3 donors. Those were not available, though, to us for in vivo studies and we were limited to this one blood type ABO rhesus macaque described in this manuscript.

A final point, these observations were borne out of the goal of establishing a NHP model in which to test HIP therapeutics. In doing so, the authors identified yet another key difference in immunobiology between humans and NHP. As increasingly sophisticated and targeted therapies are developed for use in humans, the ability to test these therapies in animal models will become increasingly difficult. In xenotransplantation, recent attention has turned to the use of human decedents as a model in which xenografts or other therapeutics could be tested in a human, at least for a short period of time.

Despite these limitations, the underlying observation remains important to multiple fields and reveals another layer of complexity in our understanding of innate immunity. I applaud the authors for their work.

We thank the reviewer very much for their appreciation of our work.

Minor/Typographic Points:

2.4 - *Figure 4A caption "Primar"*

Thank you very much. This typo was corrected.

2.5 - *Supplemental Figure 1 caption: the panels are incorrectly referenced in the caption as "A" and "B" rather than "A", "B", and "C"*

The figure legend was corrected.

2.6 - *Supplemental Figure 3: the caption refers to panels A, B, C, and D while the figure is labeled H, I, J, and K.*

This figure legend was also corrected. Thank you very much for noticing.

2.7 - *In the methods, a few portions are duplicated:*

---- *lines 512-515 and 518-521*

---- *lines 526-533 and 542-547*

I don't see these duplications.

Reviewer #3 (Remarks to the Author):

This is an important study that experimentally interrogates the role of polymorphonuclear cells (PMNs) in allojection of wild type and hypimmune gene edited pluripotent stem cell (PSC)-derived cell therapies. PMNs play an established role in transplant allojection, however, they are not well-studied in the context of PSC regenerative medicine therapies. This manuscript has the potential to add important insights to the field with regard to PMNs in the context of xenotransplantation and regenerative medicine. However, there are significant weaknesses that need to be addressed prior to publication.

3.1 *It is unclear why there is a need to study the non-human primate (NHP) immune responses to human therapies, and why additional NHP-specific gene edits (e.g., macaque CD47) should be edited into human PSCs. Human cell transplant studies have been conducted in NHPs, and NHP-to-NHP studies have been conducted to mimic the robust patient physiology and immune responses, with the goal of translating the findings to the clinic. Addition of NHP-specific edits into human cells is*

not clinically relevant, and the practical value and even basic science understanding aspect of advancing these mechanistic studies is unclear. This should be justified much more clearly.

We appreciate the reviewer's feedback and have revised the structure of the manuscript. We use the NHP model with human HIP* cells as in vivo model to study PMN killing and pharmacologic inhibition. From there on, we move towards the engineering of pig cells for xenotransplantation into humans.

3.2 One thing to consider is that this manuscript could be split into two separate manuscripts--one focused on human into NHP transplants and the other looking at CD200 and CD99 overexpression to overcome PMN rejection responses in the setting of pig xenotransplantation into humans. There is a tie-in with the PMNs in both studies, but the latter is more clinically relevant to pig xenotransplantation—a very intriguing application of this technology. Consider separating this into two manuscripts, where each section could be more-fully developed, otherwise the connection between the two should more clearly explained.

That was a valuable comment by the reviewer, and we have revised the structure of the manuscript. We use the human-to-NHP xenograft model for which we have historical immune data to dig in further on the issues of xenograft rejection in vivo. We then use the translationally relevant pig-to-human model for in vitro studies.

3.3 Assuming the manuscript will proceed as written, PMNs should be discussed in greater detail. Are neutrophils the key PMN mediator of these responses or are other granulocyte cell types involved? This should be discussed and at minimum there should be profiling (flow cytometry) of the PMNs to show the various subsets, and to see how these vary between individual monkeys (Fig 3) and humans (Fig 4). If it is not possible to get samples from the exact animals and human donors for these experiments, multiple replicates should be included from representative samples to show the readers the composition of the PMNs and how they vary between individuals.

We have performed the experiments the reviewer suggested. We used 5 human PMN donors and 5 rhesus PMN donors and performed phenotyping of PMN subpopulations (see **Supplementary Fig. 5**).

3.4 The development of teratomas is a significant safety concern. Any potential cell therapy will need to be characterized for phenotype and purity. This information is lacking from the manuscript, and the formation of teratomas illustrates why this important information is needed. Please add methods describing the characterization done in your experiments, and data (in figure, table, or text form) illustrating the purity of the cells used in all experiments. This could be added in the methods (e.g., “for these experiments ECs are classified as CD31+ CD45- etc cells, experiments used 90% +/- 10% pure EC cells...”)

With immunology in the focus, we did not optimize our differentiation protocol into iECs and did not include any purification steps. Thus, we did have residual cells that did not express endothelial cell markers. We have added this data in **Supplementary Fig. 4**.

3.5 An additional experiment should be added to Figure 4 to assess whether CD200 and CD99 overexpression in pig ECs has a protective effect against not just human PMNs, but human PBMCs. The receptor for CD200 is expressed not only on neutrophils but also macrophages, and some T cells and B cells, so there may be a benefit above PMNs extending to other effector immune cell types.

We performed the experiments the reviewer was suggesting and included pig HIP* cells also expressing CD99 and CD200. Overall, CD99 and CD200 were not strong enough to prevent PBMC-mediated rejection of pig ECs. The data are presented in **Fig. 6d-f**.

3.6 The abstract states, in reference to Figure 2, that “macaque PMN response was...not observed with macaque EC allografts in vitro.” However, that is not completely accurate since a response was

shown in the (arguably more-clinically relevant) conditions of +DAMPs and +IL2 (both of which would be present in the inflammatory post-surgical environment encountered by various types of PSC grafts). This should be clarified.

Thank you very much for picking that up. The wording was adjusted.

3.7 An additional experiment should be conducted, or at least discussed, where CD200 and CD99 overexpression is added to NHP cells and put into the in vitro assay with NHP PMNs + DAMPS and + IL2. The authors' very encouraging finding with the pig cells would suggest that these additional edits could confer protection in the 100% NHP setting (i.e., edited NHP PSC-ECs and NHP PMNs). Further, the entirely NHP study would have direct clinical relevance owing to NHPs well-established use in pre-clinical pipelines. The current NHP vs human xeno focused study seems to miss an opportunity to study these promising new edits in an NHP model system that could be used to justify human clinical trials.

We performed the experiments the reviewer was suggesting and expressed CD99 and CD200 on rhesus wt ECs. Indeed, rhesus wt.99.200 ECs were protected from rhesus PMNs, even when stimulated with DAMPs or macaque IL-2. The data are presented in Fig. **6a-b**.

3.8 This manuscript is important because of the focus on PMNs, including neutrophils. However, authors state that PMNs are "not usually associated with graft rejection" yet there is a substantial body of literature describing the important role that neutrophils play in solid organ transplant rejection. The text should be modified to correct this error and to include relevant references (e.g. <https://doi.org/10.1038/s41423-023-01058-1> ; doi: 10.1111/ajt.13940 ; <https://doi.org/10.1016/j.trim.2023.101898>). Additionally, there is at least one other report of evaluation of neutrophils in the context of PSC hypoimmune therapies (doi: 10.1101/2024.06.07.597791)—this should be cited as it highlights the authors' case that this is an important and growing area of research relevant to regenerative medicine.

We want to thank the reviewer for making this important point and helping us focus on better describing the current knowledge on PMNs in transplantation. We have changed the sentence to "...not often assessed in transplant studies" to make the point that PMN studies are still not "mainstream" and rarely performed in transplant studies. However, as the reviewer has rightly stated, there is a growing body of evidence that PMNs play a critical role in transplantation. We have added a full paragraph in the **Discussion** to summarize PMN involvement in graft injury and rejection.

3.9 It is unclear from the figure legend in Figure 2 why the scale for in vitro tests goes to 125% instead of 100% and why one sample in B is offscale. Please clarify.

The percent drop in cell index (CI) at 90 h in impedance cytotoxicity assays is shown in these graphs as it approximates cell survival or death. Ideally, values should range between 0% and 100% killing. However, values below 0 and above 100 can occur. Negative values reflect an increase in CI from proliferation of the plated cells or change in their cell morphology that enhances coverage of the electrodes of the specialized e-plates. Values > 100 reflect increased conductivity from cellular electrolytes after complete disintegration of the target cells. Most values in this study fell into the range of -25% to 125% except for a few data points. We did not want to change the scale to maintain comparability across graphs. However, we have now extended the y-axes to cover these data points. This topic is now also better described in the **Methods**.

3.10 The role of ABO mismatch/match should be discussed in more detail.

We have expanded the discussion around ABO matching in regenerative cell therapy in the **Discussion**. Overall, ABO matching has not yet been widely accepted for the transplantation of cell therapeutics.

Overall, the reviewers' suggestions have been very helpful and guided us to markedly improve our manuscript. I will be happy to answer additional questions any time.

We would be very happy if the editors would accept this thoroughly revised manuscript for publication as an Article in *Nature Communications*.

I am looking forward to hearing from you. Please do not hesitate to contact me in case of further questions.

Sincerely,

Sonja

Dear Reviewers:

Thank you very much for your second review of our manuscript **NCOMMS-24-62524A** for publication in Nature Communications.

We appreciate the additional comments made by the Editor and the Reviewers. Please find a detailed point-by-point response to the comments (*in blue italics*) below.

Reviewer #1 (Remarks to the Author):

I have to admit that the authors tried to satisfy the reviewers and put a lot of work and effort in the revision and point to point reply. As for some concerns of the reviewers they did a great job, however, for other parts of the manuscript it became even more confusing. The manuscript was rewritten to consist of 2 parts like suggested by another reviewer, which increases my concerns about the first part of the manuscript: NHP-specific edits into human cells is for human-to-NHP xenoTX studies not clinically relevant. The importance of PMN is undisputed, but showing this in the translationally relevant pig-to-human model would have been more interesting for the (small) XenoTX community.

We thank Reviewer 1 for their feedback and would like to provide more rationale for the studies we have performed. The pig-to-human model is undoubtedly most clinically relevant, and we have taken the advice from all reviewers during the last revision very seriously and have added more data to strengthen this aspect. We went even beyond what was requested and generated pig HIP* ECs expressing human CD99 and CD200 and showed their immune evasiveness from PBMCs and PMNs (**Fig. 6**). We are very happy about your guidance in this respect.

The human-to-NHP studies are still relevant for the field because human cell products are often being tested in NHPs given that primates are still the closest relative to humans before entering clinical trials. The fact that human cells are susceptible to NHP PMN killing is very relevant for the interpretation of data and the lack of engraftment in NHPs does not predict how the human cell product would engraft in humans.

The authors state that “We thus do believe that this publication will help promote the awareness of ABO being important for cell transplantation.” There is a huge amount of data and expertise on the topic of ABO matching for HSCT, islet cell transplants, ..., the concept of necessity for matching has also been challenged in solid organ transplantation (e.g. Lori West in pediatric HTx, enzymatic conversion of blood groups of human kidneys, ...). The fact that NHP centers don't do it on a regular basis does not preclude the importance of the concept of incompatibility that is, like in humans, elegantly shown by the authors in new fig.S1A.

We fully agree with your assessment of the importance of ABO matching in clinical stem cell and solid organ transplantation and are well aware of clinical practices and the literature around it. Lori West, a personal friend, was among the first in the 1990s to perform intentional ABO-incompatible heart transplants in infants, who did not yet produce major blood-group antibodies (West LJ. *N Engl J Med.* 2001;344:793-800). This strategy has been adopted by many other centers and ABO-incompatible heart transplants in infants are now standard. A recent Pediatric Heart Transplant Society registry study found that post-transplant survival was similar despite children receiving ABO-incompatible heart transplant having a higher risk profile (Urschel S. *Lancet Child Adolesc Health.* 2021;5:341-349). Two successful ABO-incompatible heart transplants have been reported in selected patients with particularly low anti-A and anti-B titers (Tydén G. *J Heart Lung Transplant.* 2012;31:1307-10). However, an International Society for Heart and Lung Transplantation registry analysis of 94 accidental ABO-incompatible heart transplants in adults revealed elevated early graft failure and mortality (Bergenfeldt H. *J Heart Lung Transplant.* 2015;34:892-8). In summary, ABO compatibility is an important factor for successful heart transplantation. For kidney transplantation, a protocol enabling safe ABO-

incompatible transplantation was developed that included the antigen-specific immunoadsorption to remove existing anti-A or anti-B antibodies and rituximab to prevent a rebound of blood type antibodies at later time points (Tydén G. *Transplant Proc.* 2005;37:3286-7). This preconditioning was combined with traditional immunosuppression using tacrolimus, mycophenolate mofetil, and prednisolone and resulted in ABO-incompatible kidney transplants having graft survival equal to ABO-compatible living-donor renal transplants (Tydén G. *Transplantation.* 2007;83:1153-5). These data further support the importance of blood group antigens in transplantation, as Reviewer 1 points out. We are in full agreement.

The fact that the primate research centers we approached did not perform ABO testing does not in any way minimize the relevance of blood group antigens. It points towards an unawareness of the relevance of blood group matching when human cells are injected into NHPs and to the fact that it is not routinely performed in research settings.

The number of animals as well as information about the sex in the in vivo experiments are now mentioned in the Methods. The authors also added number of animals in the Figure legends. Even though this was what I requested, the fact that Fig.2 is dedicated to the results of ONE single animal is something that (at least in my personal opinion and I am really kind of sorry for that) makes the reader suspicious and does not help the credibility of the data or manuscript.

We are sorry for the confusion around animal numbers and would like to provide more clarity. All conclusions drawn in this paper have been derived from at least 3 animals per group for in vivo experiments or at least 3 different effector cell donors for in vitro experiments.

Fig. 2 describes the pivotal experiment that pointed us towards looking into NHP PMNs as effector cells in cell transplantation experiments. As we describe in detail in the manuscript, we were only able to include one rhesus monkey with blood type AB that was available and compatible with our human cell product that has blood group A. We found that the only effector cell population reactive against our human HIP* iEC grafts was NHP PMNs. Since we were not able to find enough rhesus monkeys with compatible blood type to run studies with adequate group sizes, we switched to cynomolgus monkeys for the subsequent in vivo experiments. No conclusion was drawn from this Figure. It provides the rationale for looking into PMN involvement in xenotransplantation and for switching from rhesus monkeys to cynomolgus monkeys.

Fig. 3 shows the mechanisms by which allogeneic or xenogeneic PMNs get activated and kill their targets or get inhibited and spare their targets. Three different effector cell donors have been used for all graphs and all individual data points are shown.

Fig. 4 shows the intramuscular transplants of human HIP* iECs in 4 cynomolgus monkeys. We explanted one graft after 10 days for histology and followed the 3 remaining animals in this group using BLI imaging to assess graft survival.

Fig. 5 similarly shows the intramuscular transplants of human wt iECs in another 4 cynomolgus monkeys. Again, we explanted one graft after 10 days for histology and followed the 3 remaining animals in this group using BLI imaging to assess graft survival.

Fig. 6 shows allogeneic rhesus experiments and xenogeneic pig-to-human in vitro experiments. We used 3 different rhesus PMN donors and 5 different human PMN and PBMC donors. All individual data points are shown.

In summary, all in vivo group sizes were at least 3 animals and at least 3 different effector cell donors were used for all in vitro experiments.

Minor point: the title does only cover for one part of the work now.

Thank you very much for this comment. We thought long and hard about the best possible title and wanted to make sure it captures the main takeaway point of the manuscript. We think this title does that and journal guidelines prevent the creation of lengthy titles. We hope we were able to find an appealing compromise.

Although interesting from a basic science perspective (and without limiting the authors credibility or scientific impact and importance), I still have major concerns that this manuscript provides robust data that allow generalized conclusions about the use of HIP cells between different species and the impact for the field of xenotransplantation.

We hope the above clarifications were helpful in resolving the remaining concerns.

Reviewer #2 (Remarks to the Author):

I appreciate the authors thoughtful responses to all reviewers comments. I believe all of the comments and critiques have been adequately addressed. The manuscript has been strengthened and the results are more clearly presented. Thank you for the opportunity to review this work.

We want to thank this reviewer for the guidance that was provided and the very constructive comments that have helped to make this manuscript better.

Reviewer #3 (Remarks to the Author):

The revised manuscript has meaningfully improved compared to the original submission. The comments to the reviewers were near satisfactory but multiple responses need further clarification. Most importantly, in multiple instances the authors state that they have revised the manuscript in light of specific reviewer suggestions but no page/line #s are given so it cannot be verified whether satisfactory changes have been made or not. For every mention of the manuscript being revised, please provide specific page and/or line numbers (see below for more details). Please address the following follow-up questions related to specific author responses.

3.1: Please point out the specific parts of the manuscript structure that was revised.

We are happy to provide better details and have included line numbers throughout the manuscript to be able to direct the reviewer more precisely.

On **why** the human cells have to be engineered to be compatible with NHP immunology is mentioned on page 2 line 30: “Transgenes expressed on cellular grafts must be of the recipient species to properly interact with the recipient immune system or have to be sufficiently cross-reactive to be functionally competent.” In the following paragraph we detail the incompatibility of the CD47-SIRP α axis between humans and macaques.

If human cell products are tested in NHPs because they are still the most closely related non-human species, then they must be compatible for the specific function that is being tested. In our case, we want to assess immune evasion. It would be less useful to knowingly transplant cells with known incompatibility in the critical pathway that makes the HIP cell evade immune rejection. Therefore, we had to make HIP* iECs to allow testing in macaques.

To improve the flow of the manuscript, we make a similar comment about the pig-to-human model on page 2, line 40: “Similarly, for pig-to-human xenotransplant studies, we used pig HIP* ECs with depleted swine leukocyte antigen (SLA) class I and II and human CD47 transgene expression (**Fig. 1c**)”.

For the same reason as above, pigs or pig cells engineered for xenotransplantation into humans have to express human transgenes. We made a new Fig. 1 that illustrates the engineering of human HIP* cells for xenotransplantation into NHPs and of pig HIP* ECs for transplant experiments with human immune cells.

3.2: Please point out the specific parts of the manuscript structure that was revised.

In an effort to tie the NHP in vivo experiments and the pig experiments more closely together, we have created Fig. 1, which shows both engineered human cells and engineered pig cells.

The paragraph starting on page 3, line 1 summarizes the scope of the manuscript: "This study reveals that polymorphonuclear cells (PMNs) pose an additional immune barrier for the xenotransplantation of HIP cells. We show that pharmacologic inhibition of PMNs or the overexpression of the PMN-inhibitory ligands CD99 and CD200 prevent PMN cytotoxicity. Pig HIP* cells expressing CD99 and CD200 became fully protected against all human adaptive and innate immune cell cytotoxicity including PMNs".

The whole last paragraph in the Results section starting on page 6, line 16 ties NHP PMN experiments to the subsequent experiments with engineered pig cells: "CD99 and CD200 are inhibitory ligands for the receptors CD200R and PIRL α on immune cells including PMNs (**Fig. 6a**). We initially expressed CD99 and CD200 on rhesus wt ECs (rhesus wt.99.200 ECs, **Supplementary Fig. 3a**) for the testing in an allogeneic setting with rhesus PMNs. Indeed, this engineering effectively prevented the killing of wt.99.200 ECs by allogeneic rhesus PMNs activated with DAMPs or macaque IL-2 (**Fig. 6b**). In a next step, this engineering strategy was tested in a translationally highly relevant xenotransplant model. Clinical xenotransplantation of hearts and kidneys from genetically engineered pigs into patients not eligible for allotransplantation have recently been performed. We thus aimed to assess whether human PMNs would exert xeno-reactive cytotoxicity against pig wt ECs, and if so, whether CD99 and CD200 expression could inhibit that (**Fig. 6c, Supplementary Fig. 3b-d**). Using PMNs from 5 healthy volunteers, we could see cytotoxicity against pig wt ECs even without PMN stimulation (**Fig. 6d**). Primed PBMCs from 5 healthy volunteers expectedly killed the pig wt ECs in what would be considered adaptive xenorejection. The engineered pig wt.99.200 ECs were successfully protected from activated human PMNs but were still rejected by primed human PBMCs (**Fig. 6e**). To fully overcome xenorejection, pig HIP*.99.200 ECs were engineered (**Fig. 6c**). Indeed, pig HIP*.99.200 ECs were fully protected from activated human PMNs as well as from primed human PBMCs in vitro (**Fig. 6f**)".

We hope this shows how the manuscript was restructured to tie NHP and pig experiments together in a better way. We thank the reviewer for this helpful suggestion.

3.3: This figure is very helpful. Please update it further to add percentages in the all gates (currently, you only show percentages for the eosinophils, and in the bar graphs). The figure should show the percents of each cell type, since it is difficult to determine exactly from the bar graphs. Given that the vast majority of your PMNs are neutrophils, this is now appropriately emphasized in the manuscript Discussion. It is helpful to discuss specific cells within the PMN population, as each cell type plays a unique role and their frequencies vary considerably within a given blood sample i.e., neutrophils comprise the vast majority of PMNs.

Percentages were added to all gates in **Supplementary Fig. 5**. Also, to improve the reading of the bar graphs, we have broken the y-axis into two segments and zoom in on the smaller numbers. The percentages for eosinophils and basophils can now be determined more easily.

3.4: I disagree with the authors' assessment that immunology studies should not include analysis of

cell purity. Graft composition, especially with regards to the purity of the cells within, is helpful and important information in the context of immunology. Arguably, teratoma formation due to presence of undifferentiated PSCs is MORE of a concern in the context of hypoimmune cell therapies because of the potential for unchecked growth of now-hypoimmune tumors. Additionally, prior research has shown that there can be significant differences in how the immune system responds to PSCs vs PSC-differentiated cells from the same donor. See PMID: 23371903 for more information regarding immunogenicity of PSCs vs their differentiated products. Please cite this manuscript and also clarify in the methods whether the purity shown in the Supplementary Figure 4 is representative of a typical differentiation, or if that varies and it is possible that your manuscript contains impure cell populations. Highly-pure cell preparations will be a requirement for any cellular therapy (i.e., an overarching goal of this work).

Thank you very much for this important comment. We have stated in the Methods that the shown plots are representative. On page 25, line 19: “**Supplementary Fig. 4** shows representative flow cytometry plots”.

The publication by DeAlmeida P et al. has been included and we now bring up the topic of differences in immune response towards iPSCs and their derivatives. On page 5, line 42: “The lower level of immunogenicity of pluripotent stem cells might have been permissive for teratoma growth in both groups. Differences in the immune response against iPSCs and their derivatives have to be considered in this context.”

3.6: Please note which lines were adjusted.

The Summary has been adjusted so that the statement that macaque PMN response was not observed with macaque EC allografts in vitro has now been removed.

All other questions were addressed satisfactorily.

Thank you very much.

We would be very happy if the editors and reviewers would accept this thoroughly revised manuscript for publication as an Article in *Nature Communications*.

I am looking forward to hearing from you. Please do not hesitate to contact me in case of further questions.

Sincerely,

Sonja

Dear Reviewers

Thank you very much for reviewing our response for our manuscript “Inhibition of polymorphonuclear cells averts cytotoxicity against hypimmune cells in xenotransplantation” **NCOMMS-24-62524B**. And thank you for in principle acceptance.

We have made all the Editorial revisions and revisions suggested by the Reviewers. Please find a detailed point-by-point response to the comments (*in blue italics*) below.

Reviewer #1 (Remarks to the Author):

In the revised version the authors provided improved rationale for the studies with regard to the clinical relevance of the pig-to-human model. Moreover, the point to point reply contains a very interesting summary of the rationale and importance of ABO matching (which might also be interesting for others and could (in part?) be included into the manuscript?

Thank you for clarification on the animal numbers which in my opinion is crucial in preclinical animal studies.

In summary this version is significantly improved compared to the initial manuscript.

Great point. We have now included a shortened summary of why ABO matching is important for solid organ transplantation and what needs to be done to successfully perform ABO-incompatible transplants (page 7, line3).

Reviewer #3 (Remarks to the Author):

The authors have sufficiently addressed all of my comments, with one exception.

The following portion should be updated to acknowledge that complexity of their experimental model: page 2 line 30: “Transgenes expressed on cellular grafts must be of the recipient species to properly interact with the recipient immune system or have to be sufficiently crossreactive to be functionally competent.” The concept of modifying human cells to work in NHP potentially makes it less clinically relevant than had they used entirely human cells/human edits in the NHP system, as other preclinical studies have done. (see Zhu et al 2018 Circ Res; Du et al 2022 Nat Med). Making human cells work better in NHP than they otherwise would have doesn't really have an analogy in the human clinical setting. In fact, the edits described potentially introduce additional, new variables that would not be encountered in an entirely human setting. Please address this directly, in the context of your overall goal of creating human therapeutics.

Thank you for this comment. We have adapted the sentence to reflect this point (page 2, line 32).

We would be very happy if the editors would accept our final revisions for publication in Nature Communications.

Sincerely,

Sonja